# Computing Accurate Probabilistic Estimates of One-D Entropy from Equiprobable Random Samples

**DOI:** 10.3390/e23060740

**Published:** 2021-06-11

**Authors:** Hoshin V. Gupta, Mohammad Reza Ehsani, Tirthankar Roy, Maria A. Sans-Fuentes, Uwe Ehret, Ali Behrangi

**Affiliations:** 1Hydrology and Atmospheric Sciences, The University of Arizona, Tucson, AZ 85721, USA; rehsani@email.arizona.edu (M.R.E.); behrangi@email.arizona.edu (A.B.); 2Civil and Environmental Engineering, University of Nebraska-Lincoln, Omaha, NE 68182, USA; roy@unl.edu; 3GIDP Statistics and Data Science, The University of Arizona, Tucson, AZ 85721, USA; sans@email.arizona.edu; 4Institute of Water and River Basin Management, Karlsruhe Institute of Technology (KIT), 76131 Karlsruhe, Germany; uwe.ehret@kit.edu

**Keywords:** entropy, estimation, quantile spacing, accuracy, uncertainty, bootstrap, small-sample efficiency

## Abstract

We develop a simple Quantile Spacing (QS) method for accurate probabilistic estimation of one-dimensional entropy from equiprobable random samples, and compare it with the popular Bin-Counting (BC) and Kernel Density (KD) methods. In contrast to BC, which uses equal-width bins with varying probability mass, the QS method uses estimates of the quantiles that divide the support of the data generating probability density function (pdf) into equal-probability-mass intervals. And, whereas BC and KD each require optimal tuning of a hyper-parameter whose value varies with sample size and shape of the pdf, QS only requires specification of the number of quantiles to be used. Results indicate, for the class of distributions tested, that the optimal number of quantiles is a *fixed fraction* of the sample size (empirically determined to be ~0.25–0.35), and that this value is relatively insensitive to distributional form or sample size. This provides a clear advantage over BC and KD since hyper-parameter tuning is not required. Further, unlike KD, there is no need to select an appropriate kernel-type, and so QS is applicable to pdfs of arbitrary shape, including those with discontinuous slope and/or magnitude. Bootstrapping is used to approximate the sampling variability distribution of the resulting entropy estimate, and is shown to accurately reflect the true uncertainty. For the four distributional forms studied (*Gaussian*, *Log-Normal*, *Exponential* and *Bimodal Gaussian Mixture*), expected estimation bias is less than 1% and uncertainty is low even for samples of as few as 100 data points; in contrast, for KD the small sample bias can be as large as −10% and for BC as large as −50%. We speculate that estimating quantile locations, rather than bin-probabilities, results in more efficient use of the information in the data to approximate the underlying shape of an unknown data generating pdf.

## 1. Introduction

Consider a data generating process $p\left(x\right)$ from which a finite size set of $N_{S}$ random, equiprobable, independent identically distributed (iid) samples $S=\left\{s_{i},\;i=1\ldots N_{S}\right\}$ is drawn. In general, we may not know the nature and mathematical form of $p\left(x\right)$, and our goal is to compute an estimate ${\hat{H}}_{p}\left(X|S\right)$ of the Entropy $H_{p}\left(X\right)$ of $p\left(x\right)$.

In the idealized case, where X is a one-dimensional continuous random variable and the parametric mathematical form of $p\left(x\right)$ is known, we can apply the definition of *differential* Entropy [1,2] to compute:(1)$$H_{p}\left(X\right)=E_{p}\left\{-ln\left(p\left(x\right)\right)\right\}=\int_{-\infty }^{+\infty }-ln\left(p\left(x\right)\right)\cdot p\left(x\right)\cdot dx$$

Explicit, closed form solutions for $H_{p}\left(X\right)$ are available for a variety of probability density functions (pdfs). For a variety of others, closed form solutions are not available, and one can compute $H_{p}\left(X\right)$ via numerical integration of Equation (1). In all such cases, entropy estimation consists of first obtaining estimates $\hat{\theta }|S$ of the parameters θ of the known parametric density $p\left(x|\theta \right)$ and then computing the entropy estimate ${\hat{H}}_{p|\hat{\theta }}\left(X|S\right)$ by plugging $p\left(x|\hat{\theta }\right)$ into Equation (1). Any bias and uncertainty in the entropy estimate will depend on the accuracy and uncertainty of the parameter estimates $\hat{\theta }$. If the form of $p\left(x|\theta \right)$ is “*assumed*” rather than explicitly known, then additional bias will stem from the inadequacy of this assumption.

In most practical situations the mathematical form of $p\left(x\right)$ is not known, and S must first be used to obtain a data-based estimate $\hat{p}\left(x|S\right)$, from which an estimate ${\hat{H}}_{\hat{p}}\left(X|S\right)$ can be obtained via numerical integration of Equation (1). In generating $\hat{p}\left(x|S\right)$, consistency with prior knowledge regarding the nature of $p\left(x\right)$ must be ensured—for example, X may be known to take on only positive values, or values on some finite range. Consistency must also be maintained with the information in S. Further, the sample size $N_{S}$ must be sufficiently large that the information in S provides an accurate characterization of $p\left(x\right)$—in other words, that S is informationally representative and consistent.

To summarize, for the case that X is a continuous random variable, entropy estimation from data involves two steps; (i) Use of S to estimate $\hat{p}\left(x|S\right)$, and (ii) Numerical integration to compute an estimate of entropy using Equation (2):(2)$${\hat{H}}_{\hat{p}}\left(X|S\right)=E_{\hat{p}}\left\{-ln\left(\hat{p}\left(x|S\right)\right)\right\}=\int_{-\infty }^{+\infty }-ln\left(\hat{p}\left(x|S\right)\right)\cdot \hat{p}\left(x|S\right)\cdot dx$$

Accordingly, the estimate ${\hat{H}}_{\hat{p}}\left(X|S\right)$ has two potential sources of error. One is due to the use of $\hat{p}\left(x|S\right)$ to approximate $p\left(x\right)$, and the other is due to imperfect numerical integration. To maximize accuracy, we must ensure that both these errors are minimized. Further, ${\hat{H}}_{\hat{p}}\left(X|S\right)$ is a statistic that is subject to inherent random variability associated with the sample S, and so it will be useful to have an uncertainty estimate, in some form such as confidence intervals.

For cases where X is discrete and can take on only a finite set of values $\left\{x^{\left(j\right)},j=1,\ldots N_{X}\right\}$**,** if the mathematical form of $p\left(x\right)=\left\{p\left(x^{\left(j\right)}\right),j=1,\ldots N_{X}\right\}$ is known, then $H_{p}\left(X\right)$ can be computed by applying the mathematical definition of *discrete* Entropy:(3)$$H_{p}\left(X\right)=E_{p}\left\{-ln\left(p\left(x\right)\right)\right\}=-\overset{N_{X}}{\underset{j=1}{\sum }}ln\left(\;p\left(x^{\left(j\right)}\right)\right)\cdot p\left(x^{\left(j\right)}\right)$$

Here, given a data sample S, pdf estimation amounts simply to counting the number $n^{\left(j\right)}$ of data points in S that take on the value $x^{\left(j\right)}$, and setting $\hat{p}\left(x|S\right)=\left\{\hat{p}\left(x^{\left(j\right)}|S\right)=\frac{n^{\left(j\right)}}{N_{S}},\;j=1,\ldots N_{X}\right\}$ and $\hat{p}\left(x|S\right)=0$ otherwise. Entropy estimation then consists of applying the equation:(4)$${\hat{H}}_{\hat{p}}\left(X|S\right)=-\overset{N_{X}}{\underset{j=1}{\sum }}ln\left(\;\hat{p}\left(x^{\left(j\right)}|S\right)\right)\cdot \hat{p}\left(x^{\left(j\right)}|S\right)$$

In this case, there is no numerical integration error; any bias in the estimate is entirely due to $\hat{p}\left(x^{\left(j\right)}|S\right)\neq \;p\left(x^{\left(j\right)}\right)$, which occurs due to S not being perfectly informative about $p\left(x\right)$, while uncertainty is due to S being a random sample drawn from $p\left(x\right)$. If S is a representative sample, as $N_{S}\to \infty$ then $\hat{p}\left(x^{\left(j\right)}|S\right)\to p\left(x^{\left(j\right)}\right)$ and hence ${\hat{H}}_{\hat{p}}\left(X|S\right)\to H_{p}\left(x\right)$, so that estimation bias and uncertainty will both tend towards zero as the sample size is increased.

When the one-dimensional random variable X is some hybrid combination of discrete and continuous, the relative fractions of total probability mass associated with the discrete and continuous portions of the pdf must also be estimated. The general principles discussed herein also apply to the hybrid case, and we will not consider it further in this paper; for a relevant discussion of estimating entropy for mixed discrete-continuous random variables, see [3].

## 2. Popular Approaches to Estimating Distributions from Data

We focus here on the case of a one-dimensional continuous random variable X for which the mathematical form of $p\left(x\right)$ is unknown. [4] provides a summary of methods for the estimation of pdfs from data, while [5] provides an overview of methods for estimating the differential entropy of a continuous random variable. The three most widely used “*non-parametric*” methods for estimating differential entropy by pdf approximation are the: (a) Bin-Counting (BC) or piece-wise constant frequency histogram method, (b) Kernel Density (KD) method, and (c) Average Shifted Histogram (ASH) method. Ref [6] points out that these can all be asymptotically viewed as “*Kernel*” methods, where the bins in the BC and ASH approaches are understood as treating the data points falling in each bin as being from a locally uniform distribution. For a theoretical discussion of the basis for non-parameteric estimation of a density function see also [7].

As discussed by [8,9], appropriate selection of the bin-width (effectively a smoothing hyper-parameter) is critical to success of the BC and ASH histogram-based methods. Bin-widths that are too small can result in overly rough approximation of the underlying distribution (increasing the variance), while bin-widths that are overly large can result in an overly smooth approximation (introducing bias). Therefore, one typically has to choose values that balance variance and bias errors. [8] and [10] present expressions for “*optimal*” bin width when using BC, including the “*normal reference rule*” that is applicable when the pdf is approximately Gaussian, and the “*oversmoothed bandwidth rule*” that places an upper-bound on the bin-width. Similarly, [6] shows that while KD is more computationally costly to implement than BC, its accuracy and convergence are better, and they derive optimal values for the KD smoothing hyper-parameter. [11] also proposed the ASH method, which refines BC by sub-dividing each histogram bin into sub-bins, with computational cost similar to BC and accuracy approaching that of KD. Note that, if prior information on the shape of $p\left(x\right)$ is available, or if a representation with the smallest number of bins is desired, then variable bin width methods may be more appropriate (e.g., [12,13,14,15,16]).

The BC, KD and ASH methods all require hyper-parameter tuning to be successful. BC requires selection of the histogram bin-widths (and thereby the number of bins), KD requires selection of the form of the Kernel function and tuning of its parameters, and ASH requires selection of the form of a Kernel function and tuning of the coarse bin width, and number of sub-bins. While recommendations are provided to guide the selection of these “*hyper-parameters*”, these recommendations depend on theoretical arguments based in assumptions regarding the typical underlying forms of $p\left(x\right)$. Based on empirical studies, and given that we typically do not know the “*true*” form of $p\left(x\right)$ to be used as a reference for tuning, [3] recommend use of BC and KD rather than the ASH method.

Finally, since BC effectively treats the pdf as being discrete, and therefore uses Equation (4) with each of the indices j corresponding to one of the histogram bins, the loss of entropy associated with implementing the discrete constant bin-width approximation is approximately $ln\left(\Delta \right)$**,** where Δ is the bin width, provided Δ is sufficiently small [2]. This fact allows conversion of the discrete entropy estimate to differential entropy simply by adding $ln\left(\Delta \right)$ to the discrete entropy estimate.

In summary, while BC and KD can be used to obtain accurate estimates of entropy for pdfs of arbitrary form, hyper-parameter tuning is required to ensure that good results are obtained. In the next section, we propose an alternative method to approximate $p\left(x\right)$ that does not require counting the numbers of samples in “*bins*”, and is instead based on estimating the quantile positions of $p\left(x\right)$. We first compare the properties and performance of the method with BC (rather than with KD or ASH) for a range of distributions because (i) BC is arguably the most straightforward and popular method, (ii) BC shares important similarities with the proposed approach, but also shows distinct differences worthy of discussion, and (iii) we seek good performance for a wide range of distributions. At the request of a reviewer, we further report performance of KD on the same distributions, reinforcing the findings of [17] that KD performance can depend strongly on distribution shape and sample size.

## 3. Proposed Quantile Spacing (QS) Approach

We present an approach to computing an estimate ${\hat{H}}_{p}\left(X|S\right)$ of Entropy $H_{p}\left(X\right)$ given a set of available samples S for the case where X is a one-dimensional continuous random variable and the mathematical form of the data generating process $p\left(x\right)$ is unknown. The approach is based in the assumption that $p\left(x\right)$ can be approximated as piecewise constant on the intervals between quantile locations, and consists of three steps.

### 3.1. Step 1—Assumption about Support Interval

The first step is to assume that $p\left(x\right)$ exists only on some finite support interval $\left[x_{min},x_{max}\right]$, where $x_{min}\leq \min\left\{S\right\}$ and $x_{max}\geq \max\left\{S\right\}$; i.e., we treat $p\left(x\right)$ as being 0 everywhere outside of the interval $\left[x_{min},x_{max}\right]$. Given that the true support of X may, in reality, be as extensive as $\left[-\infty ,+\infty \right]$, we allow the selection of this interval (based on prior knowledge, such as physical realism) to be as extensive as appropriate and/or necessary. However, as we show later, the impact of this selection can be quite significant and will need special attention.

### 3.2. Step 2—Assumption about Approximate Form of $p\left(X\right)$

The second step is to assume that $p\left(x\right)$ can be approximated as piecewise constant on the intervals between quantiles $Z=\left\{z_{0},z_{1},z_{2},\ldots ,z_{N_{Z}}\right\}$ associated with the $\left\{0,\frac{1}{N_{Z}},\frac{2}{N_{Z}},\ldots \frac{N_{Z}-1}{N_{Z}},1\right\}$ non-exceedance probabilities of $p\left(x\right)$, where $N_{Z}$ represents the number of quantiles, $z_{0}=x_{min}$, and $z_{N_{Z}}=x_{max}$. This corresponds to making the *minimally informative* (maximum entropy) assumption that $p\left(x\right)$ is ‘*uniform*’ over each of the quantile intervals $\left[z_{j-1},z_{j}\right]$ for $j=1,\ldots N_{Z}$, which is equivalent to assuming that the corresponding cumulative distribution function $P\left(x\right)$ is piecewise linear (i.e., increases linearly between $z_{j-1}$ and $z_{j}$).

Assuming perfect knowledge of the locations of the quantiles Z, this approximation corresponds to:(5)$$p\left(x\right)\approx \hat{p}\left(x|Z\right)=p_{j-1}^{j}=\frac{K}{\Delta_{j}}\;\mathrm{for}\;z_{j-1}\leq X<z_{j},\;j=1,\ldots N_{Z}$$
where $\Delta_{j}=z_{j}-z_{j-1}$. To ensure that $\hat{p}\left(x|Z\right)$ integrates to 1.0 over the support region $\left[x_{min},x_{max}\right]$ we have $K=\frac{1}{N_{Z}}$. Accordingly, our entropy estimate is given by:(6)$${\hat{H}}_{\hat{p}}\left(X|Z\right)=\sum_{j=1}^{N_{Z}}\int_{z_{j-1}}^{z_{j}}-ln\left(p_{j-1}^{j}\right)\cdot p_{j-1}^{j}\cdot dx$$
(7)$$=\frac{1}{N_{Z}}\cdot \overset{N_{Z}}{\underset{j=1}{\sum }}ln\left(N_{Z}\cdot \Delta_{j}\right)$$
(8)$$=ln\left(N_{Z}\right)+\frac{1}{N_{Z}}\cdot \overset{N_{Z}}{\underset{j=1}{\sum }}ln\left(\Delta_{j}\right)$$

From Equation (8) we see that the estimate depends on the logs of the spacings between quantiles, and is defined by the average of these values. Further, we can define the error due to piecewise constant approximation of $p\left(x\right)$ as $\Delta H_{p,\hat{p}}\left(X|Z\right)={\hat{H}}_{\hat{p}}\left(X|Z\right)-H_{p}\left(X\right)$.

### 3.3. Step 3—Estimation of the Quantiles of $p\left(x\right)$

The third step is to use the available data S to compute estimates of the quantiles Z to be plugged into Equation (6). Of course, given a finite sample size $N_{S}$, the number of quantiles $N_{Z}$ that can be estimated will, in general, be much smaller than the sample size $N_{S}$ (i.e., $N_{Z}\ll N_{S}$).

Various methods for computing estimates of the quantiles are available. Here, we use a relatively simple approach in which $N_{K}$ sample subsets $S_{k},\;k=1,\ldots ,N_{K}$, each of size $N_{Z}-1$ (i.e., $S_{k}=\left\{s_{1}^{k},s_{2}^{k},\ldots ,s_{N_{Z}-1}^{k}\right\}$) are drawn from the available sample set S, where the samples in each subset are drawn from S without replacement so that the values obtained in each subset are unique (not-repeated). For each subset, we sort the values in increasing order to obtain $Z_{k}=\left\{z_{1}^{k},z_{2}^{k},\ldots ,z_{N_{Z}-1}^{k}\right\}$, thereby obtaining $N_{K}$ estimates $\left\{z_{j}^{1},z_{j}^{2},\ldots ,z_{j}^{N_{K}}\right\}$ for each $z_{j},\;j=1,\ldots N_{Z}-1$. This procedure results in an empirical estimate of the sample distribution $p\left(z_{j}|S\right)$ for each quantile $z_{j},\;j=1,\ldots N_{Z}-1$. Finally, we compute ${\hat{z}}_{j}=\frac{1}{N_{K}}\sum_{k=1}^{N_{K}}z_{j}^{k}$, and set $\hat{Z}=\left\{x_{min},{\hat{z}}_{1},{\hat{z}}_{2},\ldots ,{\hat{z}}_{N_{Z}-1},x_{max}\right\}$. Plugging these values into Equations (7) and (8), we get:(9)$${\hat{H}}_{\hat{p}}\left(X|\hat{Z}\right)=\frac{1}{N_{Z}}\cdot \overset{N_{Z}}{\underset{j=1}{\sum }}ln\left(N_{Z}\cdot {\hat{\Delta }}_{j}\right)$$
(10)$$=ln\left(N_{Z}\right)+\frac{1}{N_{Z}}\cdot \overset{N_{Z}}{\underset{j=1}{\sum }}ln\left({\hat{\Delta }}_{j}\right)$$
where ${\hat{\Delta }}_{j}={\hat{z}}_{j}-{\hat{z}}_{j-1}$. For practical computation, to avoid numerical problems as $N_{Z}$ becomes large so that ${\hat{\Delta }}_{j}$ becomes very small and $ln\left({\hat{\Delta }}_{j}\right)$ approaches −∞, we will actually use Equation (9). Further, we define the additional error due purely to imperfect quantile estimation to be $\Delta H_{\hat{p}}\left(X|\hat{Z},Z\right)={\hat{H}}_{\hat{p}}\left(X|\hat{Z}\right)-H_{\hat{p}}\left(X|Z\right)$.

### 3.4. Random Variability Associated with the QS-Based Entropy Estimate

Given that the quantile spacing estimates $\left\{{\hat{\Delta }}_{j},j=1,\ldots ,N_{Z}\right\}$ are subject to random sampling variability associated with (i) the sampling of S from $p\left(x\right)$, and (ii) estimation of the quantile positions ${\hat{z}}_{j}$, the entropy estimate ${\hat{H}}_{\hat{p}}\left(X|\hat{Z}\right)$ will also be subject to random sampling variability. As shown later, we can generate probabilistic estimates of the nature and size of this error from the empirical estimates of $p\left(z_{j}|S\right)$ obtained in Step 3, and by bootstrapping on S.

## 4. Properties of the Proposed Approach

The accuracy of the estimate ${\hat{H}}_{\hat{p}}\left(X|\hat{Z}\right)$ obtained using the QS method outlined above depends on the following four assumptions, each of which we discuss below:

(i)**A1:** The piecewise constant approximation $\hat{p}\left(x|Z\right)$ of $p\left(x\right)$ on the intervals between the quantile positions is adequate(ii)**A2:** The quantile positions $Z=\left\{z_{0},z_{1},z_{2},\ldots ,z_{N_{Z}}\right\}$ of $p\left(x\right)$ have been estimated accurately.(iii)**A3:** The pdf $p\left(x\right)$ exists only on the support interval $\left[x_{min},x_{max}\right]$, which has been properly chosen(iv)**A4:** The sample set S is consistent, representative and sufficiently informative about the underlying nature of $p\left(x\right)$

### 4.1. Implications of the Piecewise Constant Assumption

Assume that $\hat{p}\left(x|Z\right)$ provides a piecewise-constant estimate of $p\left(x\right)$ and that the quantile positions $Z=\left\{z_{0},z_{1},z_{2},\ldots ,z_{N_{Z}}\right\}$ associated with a given choice for $N_{Z}$ are perfectly known. Since the continuous shape of the cumulative distribution function (cdf) $P\left(x\right)$ can be approximated to an arbitrary degree of accuracy by a sufficient number of piecewise linear segments, we will have $\hat{P}\left(x|Z\right)\to P\left(x\right)$ as $N_{Z}\to \infty$.

However, an insufficiently accurate approximation will result in a pdf estimate that is not sufficiently smooth, so that the entropy estimate will be biased. This bias will, in general, be positive (overestimation) because the piecewise-constant form $\hat{p}\left(x|Z\right)$ used to approximate $p\left(x\right)$ will always be shifted slightly in the direction of larger entropy; i.e., each piecewise constant segment in $\hat{p}\left(x|Z\right)$ is a maximum-entropy (uniform distribution) approximation of the corresponding segment of $p\left(x\right)$. However, the bias can be reduced and made arbitrarily small by increasing $N_{Z}$ until the *Kullback-Leibler* divergence between $\hat{p}\left(x|Z\right)$ and $p\left(x\right)$ is so small that the information loss associated with use of $\hat{p}\left(x|Z\right)$ in place of $p\left(x\right)$ in Equation (1) is negligible.

The left panel of Figure 1 shows how this bias in the estimate of $H_{p}\left(X\right)$, due solely to piecewise constant approximation of the pdf (no sample data are involved), declines with increasing $N_{Z}$ for three pdf forms of varying functional complexity (*Gaussian*, *Log-Normal* and *Exponential*), each using a parameter choice such that its theoretical entropy $H_{p}\left(X\right)=1$. Also shown, for completeness, are results for the *Uniform* pdf where only one piecewise constant bin is theoretically required. Note that because $H_{p}\left(k\cdot \left(X-\mu_{x}\right)\right)=H_{p}\left(X\right)+\ln k$, the entropy can be changed to any desired value simply by rescaling on X. For these theoretical examples, the quantile positions are known exactly, and the resulting estimation bias is due *only* to the piecewise constant approximation of $p\left(x\right)$. However, since the theoretical pdfs used for this example all have infinite support, whereas the piecewise approximation requires specification of a finite support interval, for the latter we set $\left[x_{min},x_{max}\right]$ to be the theoretical locations where $P\left(z_{0}\right)=\varepsilon$ and $P\left(z_{N_{Z}}\right)=1-\varepsilon$ respectively, with ε chosen to be some sufficiently small number (we used $\varepsilon =10^{-5}$). We see empirically that bias due to the piecewise constant approximation declines to zero approximately as an exponential function of $\log N_{Z}$ so that the absolute percent bias is less than ~10% for $N_{Z}>10–30$, less than ~5% for $N_{Z}>50$, and less than ~1% for $N_{Z}>200$.

In practice, given a finite sample size $N_{S}$, our ability to increase the value of $N_{Z}$ will be constrained by the size of the sample (i.e., $N_{Z}<$ $N_{S}$). This is because when the form of $p\left(x\right)$ is unknown, the locations of the quantile positions must be estimated using the information provided by S. Further, what constitutes a sufficiently large value for $N_{Z}$ will depend the complexity of the underlying shape of $p\left(x\right)$.

### 4.2. Implications of Imperfect Quantile Position Estimation

Assume that $N_{Z}$ is large enough for the piecewise constant pdf approximation to be sufficiently accurate, but that the estimates $\left\{{\hat{z}}_{0},{\hat{z}}_{2},\ldots ,{\hat{z}}_{N_{Z}}\right\}$ of the locations of the quantiles are imperfect. Clearly, this can affect the estimate of entropy computed via Equation (9) by distorting the shape of $\hat{p}\left(x|\hat{Z}\right)$ away from $\hat{p}\left(x|Z\right)$, and therefore away from $p\left(x\right)$. Further, the uncertainty associated with the quantile estimates will translate into uncertainty associated with the estimate of entropy.

In general, as the number of quantiles $N_{Z}$ is increased, the inter-sample spacings associated with each ordered subset $Z_{k}=\left\{z_{1}^{k},z_{2}^{k},\ldots ,z_{N_{Z}-1}^{k}\right\},\;k=1,\ldots ,N_{K}$ will decrease, so that the distribution of possible locations for each quantile $z_{j}^{k},j=1,\ldots N_{Z}-1$ will progressively become more tightly constrained. This means that the bias associated with each estimated quantile ${\hat{z}}_{j}$ will reduce progressively towards zero as $N_{Z}$ is increased (constrained only by sample size $N_{S}$) and the variance of the estimate ${\hat{z}}_{j}$ will decline towards zero as the number of subsamples $N_{K}$ is increased.

Figure 2 illustrates how bias and uncertainty associated with estimates of the quantiles diminish with increasing $N_{Z}$ and $N_{K}$. Experimental results are shown for the *Log-Normal* density with μ=0 and $\sigma^{2}=0.6577$ (theoretical entropy $H_{p}\left(X\right)=1$), with the *y*-axis indicating percent error in the quantile estimates corresponding to the 90% (green), 95% (purple) and 99% (turquoise) non-exceedance probabilities. In these plots, there is no distorting effect of sample size $N_{S}$ (the sample size is effectively infinite), since when computing the estimates of the quantiles (as explained in Section 3.3) we draw subsamples of size $N_{Z}$ directly from the theoretical pdf.

The left-side plot shows, for $N_{K}=500$ subsamples, how the biases and uncertainties diminish as $N_{Z}$ is increased. The boxplots reflect uncertainty due to random sampling variability, estimated by repeating each experiment 500 times (by drawing new samples from the pdf). As expected, for smaller $N_{Z}$ the quantile location estimates tend to be negatively biased, particularly for those in the more extreme tail locations of the distribution. However, for $N_{Z}=150$ the bias associated with the 99% non-exceedance probability quantile is less than −5%, for $N_{Z}\approx 500$ the corresponding bias is less than −2%, and for $N_{Z}=1500$ it is less than −1%. The right-side plot shows, for a fixed value of $N_{Z}=1000$, how the uncertainties diminish but the biases remain relatively constant as the number of subsamples $N_{K}$ is increased. Overall, the uncertainty becomes quite small for $N_{K}>200$.

### 4.3. Implications of the Finite Support Assumption

Assume that $N_{Z}$ has been chosen large enough for the piecewise constant pdf approximation to be sufficiently accurate, and that the exact quantile positions associated with this choice for $N_{Z}$ are known. The equation for estimating entropy (Equation (9)) can be decomposed into three terms:(11)$$H_{\hat{p}}\left(X|Z\right)=\frac{1}{N_{Z}}\cdot \overset{N_{Z}}{\underset{j=1}{\sum }}ln\left(N_{Z}\cdot \Delta_{j}\right)=H_{x_{min}}^{z_{1}}+H_{z_{1}}^{z_{N_{Z}-1}}+H_{N_{Z}-1}^{x_{max}}$$
where $H_{x_{min}}^{z_{1}}=\frac{ln\left(N_{Z}\cdot \Delta_{1}\right)}{N_{Z}}$, $H_{z_{1}}^{z_{N_{Z}-1}}=\frac{1}{N_{Z}}\cdot \overset{N_{Z}-1}{\underset{j=2}{\sum }}ln\left(N_{Z}\cdot \Delta_{j}\right)$ and $H_{N_{Z}-1}^{x_{max}}=\frac{ln\left(N_{Z}\cdot \Delta_{N_{Z}}\right)}{N_{Z}}$, and where $\Delta_{j}$ indicates the true inter-quantile spacings. Only the first and last terms $H_{x_{min}}^{z_{1}}$ and $H_{N_{Z}-1}^{x_{max}}$ are affected by the choices for $x_{min}$ and $x_{max}$ through $\Delta_{1}=z_{1}-x_{min}$ and $\Delta_{N_{Z}}=x_{max}-z_{N_{Z}-1}$.

Clearly, if $p\left(x\right)$ is bounded both above and below by specific known values, then there is no issue. However, if the support of $p\left(x\right)$ is not known, or if one or both bounds can reasonably be expected to extend to ±∞ (as appropriate), then the choice for the relevant limiting value ($x_{min}$ or $x_{max}$) can significantly affect the computed value for ${\hat{H}}_{\hat{p}}$. To see this, note that the first term $H_{x_{min}}^{z_{1}}$ can be made to vary from −∞ when $\Delta_{1}=0$, to +∞ when $\Delta_{1}=\infty$, passing through zero when $\Delta_{1}=\frac{1}{N_{Z}}$; and similarly for the last term $H_{N_{Z}-1}^{x_{max}}$. Therefore, the error associated with ${\hat{H}}_{\hat{p}}$ can be made arbitrarily negatively large by choosing $\Delta_{1}$ and $\Delta_{N_{Z}}$ to be too small, or arbitrarily positively large by choosing $\Delta_{1}$ and $\Delta_{N_{Z}}$ to be too large.

In practice, when dealing with samples S from some unknown data generating process, we will often have only the samples themselves from which to infer the support of $p\left(x\right)$, and therefore can only confidently state that $x_{min}\leq \min\left\{S\right\}$ and $x_{max}\geq \max\left\{S\right\}$. One possibility could be to ignore the fractional contributions of the terms $H_{x_{min}}^{z_{1}}$ and $H_{N_{Z}-1}^{x_{max}}$ corresponding to the (unknown) portions of the pdf and instead use as our estimate $H_{\hat{p}}^{*}\left(X|Z\right)=H_{z_{1}}^{z_{N_{Z}-1}}$. This would be equivalent to setting $\Delta_{1}=\Delta_{N_{Z}}=\frac{1}{N_{Z}}$, so that $X_{min}=z_{1}-\frac{1}{N_{Z}}$ and $X_{max}=z_{N_{Z}-1}+\frac{1}{N_{Z}}$. By doing so, we would be ignoring a portion of the overall entropy associated with the pdf and can therefore expect to obtain an underestimate. However, this bias error $BE=H_{\hat{p}}\left(X|Z\right)-H_{\hat{p}}^{*}\left(X|Z\right)$ will tend to zero as $N_{Z}$ is increased.

An alternative approach, that we recommend in this paper, is to set $x_{min}=\min\left\{S\right\}$ and $x_{max}=\max\left\{S\right\}$. In this case, there will be random variability associated with the sampled values for $\min\left\{S\right\}$ and $\max\left\{S\right\}$ and so the bias in our estimate $H_{\hat{p}}^{*}\left(X|Z\right)$ can be either negative or positive. Nonetheless, this bias error BE will still tend to zero as $N_{Z}$ is increased.

Note that the percentage contributions of the entropy fractions $H_{x_{min}}^{z_{1}}$ and $H_{N_{Z}-1}^{x_{max}}$ to the total entropy $H_{p}\left(X|Z\right)$ will depend on the nature of the underlying pdf. Figure 3 illustrates this for three pdfs (*Gaussian* which has infinite extent on both sides, and the *Exponential* and *Log-Normal* which have infinite extent on only one side), assuming no estimation error associated with the quantile locations. For the *Gaussian* (blue) and *Exponential* (red) densities, the largest fractional entropy contributions are clearly from the tail regions, whereas for the *Log-Normal* (orange) density this is not so. So, the entropy fractions can be proportionally quite large or small at the extremes, depending on the form of the pdf. Nonetheless, the overall entropy fraction associated with each quantile spacing diminishes with increasing $N_{Z}.$ For the examples shown, when $N_{Z}=100$ (left plot) the maximum contributions associated with a quantile spacing are less than 6% and when $N_{Z}=1000$ (right plot) become less than 1%. This plot illustrates clearly the most important issue that must be dealt with when estimating entropy from samples.

So, on the one hand, the cumulative entropy fractions associated with the tail regions of $p\left(x\right)$ that lie beyond $\min\left\{S\right\}$ and $\max\left\{S\right\}$ are impossible to know. On the other, the individual contributions of these fractions associated with the extreme quantile spacings $\Delta_{1}$ and/or $\Delta_{N_{Z}}$ where $p\left(x\right)$ is small can be quite a bit larger than those associated with the contributions from intermediate quantile spacings. Overall, the only real way to control the estimation bias and uncertainty associated with these extreme regions is to use a sufficiently large value for $N_{Z}$ so that the relative contribution of the extreme regions is small. This will in turn, of course, be constrained by the sample size.

### 4.4. Combined Effect of the Piecewise Constant Assumption, Finite Support Assumption, and Quantile Position Estimation Using Finite Sample Sizes

In Section 4.1, we saw that the effect of the piecewise constant assumption on the QS-based estimate of entropy is *positive* bias that diminishes with increasing $N_{Z}$. Similarly, Section 4.2 showed that the biases associated with the quantiles diminish with increasing $N_{Z}$, while the corresponding uncertainties diminish with increasing $N_{K}$. As mentioned earlier, the bias in each quantile position will be towards the direction of locally higher probability mass (since more of the equiprobable random samples will tend to drawn from this region), and therefore the estimate $\hat{p}\left(x|\hat{Z}\right)$ of $p\left(x\right)$ will be distorted in the direction of having smaller “*dispersion*” (i.e., $\hat{p}\left(x|\hat{Z}\right)$ will tend to be more ‘*peaked*’ than $p\left(X\right)$), resulting in *negative* bias in the corresponding estimate of entropy. Finally, Section 4.3 discussed the implications of the finite support assumption, given that $x_{min}$ and $x_{max}$ will often not be known.

Figure 4 illustrates the combined effect of these assumptions. Here we show how the overall percentage error in the QS-based estimate of entropy varies as a function of $\alpha =\left(N_{Z}/N_{S}\right)$, where α expresses the number of quantiles $N_{Z}$ as a fraction of the sample size $N_{S}$. Sample sets of given size $N_{S}$ are drawn from the *Gaussian* (left panel), *Exponential* (middle panel) and *Log-Normal* (right panel) densities, the quantiles are estimated using the procedure discussed in Section 3.3, $x_{min}$ and $x_{max}$ are set to be the smallest and largest data values in the set (Section 4.3), and entropy is estimated using Equation (9) for different selected values of $N_{Z}$. To account for sampling variability, the results are averaged over 500 different sample sets drawn randomly from the parent density.

The plots show how percentage estimation error (bias) varies as α (and hence $N_{Z}$) changes as a fraction of sample size $N_{S}$, for different sample sizes from 100 to 5000. As might be expected, in each case when α is too small the estimation bias is positive (over-estimation) and can be quite large due to the piecewise constant approximation. However, as α is increased the estimation bias decreases rapidly, crosses zero, and becomes negative (under-estimation) due to the combined effects of quantile position estimation bias and use of the smallest and largest sample values to approximate $x_{min}$ and $x_{max}$. Most interesting is the fact that all of the curves cross zero at approximately α≈0.25–0.35, and that this location does not seem to depend strongly on the sample size or shape of the pdf. Further, the marginal cost of setting α too large is low (less than −5% for α=0.5) compared to setting α too small. Overall, the expected bias error diminishes with increasing sample size $N_{S}$ and the optimal choice for α≈0.25–0.35.

Figure 5 illustrates both the bias and uncertainty in the estimate of entropy as a function of sample size $N_{S}$ when we specified the number of quantiles $N_{Z}$ to be 25% of the sample size (i.e., α=0.25). The uncertainty intervals are due to sampling variability, estimated by drawing 500 different sample sets from the parent population. The results show that uncertainty due to sampling variability diminishes rapidly with sample size, becoming relatively small for large sample sizes.

### 4.5. Implications of Informativeness of the Data Sample

For the results shown in Figure 4 and Figure 5, we drew samples directly from $p\left(x\right)$. In practice, we must construct our entropy estimate by using a single data sample S of finite size $N_{S}$. Provided that S is a consistent and representative random sample from $p\left(x\right)$, with each element $x_{i}$ being iid, then a sufficiently large sample size $N_{S}$ should enable construction of an accurate approximation $\hat{p}\left(x|\hat{Z}\right)$ of $p\left(x\right)$ via the QS method. However, if $N_{S}$ is too small, it can (i) prevent setting a sufficiently large value for $N_{Z}$, and (ii) tend to make the sets $Z_{k}$ sub-sampled from S to be insufficiently independent for accurate estimates of the quantile positions of $p\left(x\right)$ to be obtained. The overall effect will be to prevent $\hat{p}\left(x|\hat{Z}\right)$ from approaching $p\left(x\right)$, leading to an unreliable estimate of its entropy.

Further, even if $N_{S}$ is sufficiently large for $\hat{p}\left(x|\hat{Z}\right)\to p\left(x\right)$, sampling variability associated with randomly drawing S from $p\left(x\right)$ will result in the entropy estimate $H_{\hat{p}}\left(X|\hat{Z}\right)$ being subject to statistical variability. Figure 6 shows how bootstrapped *estimates* of the uncertainty will differ from those shown in Figure 5 above, in which we drew different sample sets from the parent population. Here, each time a sample set is drawn from the parent density we draw $N_{B}=500$ bootstrap samples of the same size $N_{S}$ from that sample set, use these to obtain $N_{B}$ different estimates of the associated entropy (using α=0.25), and compute the width of the resulting inter-quartile range (IQR). We then repeat this procedure for 500 different sample sets of the same size drawn from the parent population. Figure 6 shows the ratio of the IQR obtained using bootstrapping to that of the actual IQR for different sample sizes; the boxplots represent variability due to random sampling. Here, an expected (mean) ratio value of 1.0 and small width of the boxplot is ideal, indicating that bootstrapping provides a good estimate of the uncertainty to be associated with random sampling variability. The results show that for smaller sample sizes ($N_{S}<500$) there is a tendency to overestimate the width of the inter-quartile range, but that this slight positive bias disappears for larger sample sizes.

### 4.6. Summary of Properties of the Proposed Quantile Spacing Approach

To summarize, bias in the estimate ${\hat{H}}_{\hat{p}}\left(X|\hat{Z}\right)$ can arise due to: (a) inadequacy of the piece-wise approximation of $p\left(x\right)$, (b) imperfect estimation of the quantile positions, (c) imperfect knowledge of the support interval, and (d) the sample S not being consistent, representative and sufficiently informative. Meanwhile, uncertainty in the estimate can arise due to: (a) random sampling variability associated with estimation of the quantiles, and (b) random sampling variability associated with drawing S from $p\left(x\right)$. For a given sample size $N_{S}$, and provided that the sample is consistent, representative and fully informative, the bias and uncertainty can be reduced by selecting sufficiently large values for $N_{Z}$ and $N_{K}$ (we recommend $N_{K}=500$ and $N_{Z}=0.25\cdot N_{S}$), while the overall statistical variability associated with the estimate can be estimated by bootstrapping from S.

### 4.7. Algorithm for Estimating Entropy via the Quantile Spacing Approach

Given a sample set S of size $N_{S}$
(1)Set $x_{min}=\min\left\{S\right\}$ and $x_{max}=\max\left\{S\right\}$(2)Select values for $\psi =\left\{N_{Z},N_{K},N_{B}\right\}$. Recommended default values are $N_{Z}=\alpha \cdot N_{S}$, $N_{K}=500$ and $N_{B}=500$, with α=0.25.(3)Bootstrap a sample set $S_{b}$ of size $N_{S}$ from S with replacement.(4)Compute the entropy estimate ${\hat{H}}_{\hat{p}}\left(X|{\hat{Z}}_{b}\right)$ using Equation (9) and the procedure outlined in Section 3.(5)Repeat the above steps $N_{B}$ times to generate the bootstrapped distribution of ${\hat{H}}_{\hat{p}}\left(X|{\hat{Z}}_{b}\right)$ as an empirical probabilistic estimate $p\left({\hat{H}}_{\hat{p}}\left(X|S\right)\right)$ of the Entropy $H_{p}\left(X\right)$ of $p\left(x\right)$ given S.

## 5. Relationship to the Bin Counting Approach

Because the proposed QS approach employs a piecewise constant approximation of $p\left(x\right)$, there are obvious similarities to BC. However, there are also clear differences. First, while BC typically employs equal-width binning along the support of X, with each bin having a different fraction of the total probability mass, QS uses variable width intervals (analogous to “*bins*”) each having an identical fraction of the total probability mass (so that the intervals are wider where $p\left(X\right)$ is small, and narrower where $p\left(X\right)$ is large). Both methods require specification of the support interval $\left[x_{min},x_{max}\right]$.

Second, whereas BC requires counting samples falling within bins to estimate the probability masses associated with each bin, QS involves no “*bin-counting*”, and, instead, the data samples are used to estimate the positions of the quantiles. Since the probability mass estimates obtained by counting random samples falling within bins can be highly uncertain due to sampling variability, particularly for small sample sizes $N_{S}$, this translates into uncertainty regarding the shape of the pdf and thereby regarding its entropy. In QS, the effect of sampling variability is to consistently provide a pdf approximation that tends to be slightly more peaked that the true pdf, so that the bias in the entropy estimate tends to be slightly negative. This negative bias acts to counter the positive bias resulting from the piecewise constant approximation of the pdf.

Third, whereas BC requires selection of a bin-width hyper-parameter Δ that represents the appropriate bin-width required for “*smoothing*” to ensure an appropriate balance between bias and variance errors, QS requires selection of a hyper-parameter α that specifies the number of quantile positions $N_{Z}$ as a fraction of the sample size $N_{S}$. As seen in Figure 4, an appropriate choice for α can effectively drive estimation bias to zero, while $N_{K}$ controls the degree of uncertainty associated with the estimation of the positions of the quantiles. Further, if we desire estimates of the uncertainty in the computed value of entropy arising due to random sampling variability, we must specify the number of bootstraps $N_{B}$. In practice, the values selected for $N_{K}$ and $N_{B}$ can be made arbitrarily large, and our experiments suggest that setting $N_{K}=500$ (or larger) and $N_{B}=500$ (or larger) works well in practice. Accordingly, the QS hyper-parameter α takes the place of the BC hyper-parameter Δ in determining the accuracy of the Entropy estimate obtained from a given sample.

Our survey of the literature suggests that the problem of how to select the BC bin-width hyper-parameter Δ is not simple, and a number of different strategies have been proposed. [18] proposed to choose the number of bins based on sample size only. [8] estimated the optimal number of bins by minimizing the mean squared error between the sample histogram and the “*true*” form of the pdf (for which the shape must be assumed). [19] further developed this approach by estimating the shape of the true pdf from the interquartile range of the sample. More recently, [20] proposed a method that does not require choice of a hyper-parameter—using a Bayesian maximum likelihood approach, and assuming a piecewise-constant density model, the posterior probability for the number of bins is identified (this approach also provides uncertainty estimates for the related bin counts). Other BC approaches that provide uncertainty estimates based on the *Dirichelet*, *Multinomial*, and *Binomial* distributions are discussed by [17]. However, as shown in the next section, in practice the “*optimal*” fixed bin-width can vary significantly with shape of the pdf and with sample size.

## 6. Experimental Comparison with the Bin Counting Method

The right panel of Figure 1 shows the theoretical bias, due only to piecewise-constant approximation, associated with the estimate of $H_{p}\left(X\right)$ obtained using BC when the support interval $\left[x_{min},x_{max}\right]$ is subdivided into equal-width intervals. We can compare the results to the left panel of Figure 1 if we consider the number $N_{Bin}$ of BC bins to be analogous to the number $N_{Z}$ of spacings between quantiles for QS. Note that no random sampling variability or data informativeness issues are involved in the construction of these figures. For BC we use the theoretical fractions of probability mass associated with each of the equal-width bins, and for QS we use the theoretical quantile positions to compute the interval spacings. In both cases, to address the “*infinite support*” issue, we set $\left[x_{min},x_{max}\right]$ to the locations where $P\left(z_{0}\right)=\varepsilon$ and $P\left(z_{N_{Z}}\right)=1-\varepsilon$ respectively, with ε=1 × 10^−5^. As with QS, the bias in entropy computed using the BC equal bin-width piecewise-continuous approximation declines to zero with increasing numbers of bins, and becomes less that 1% when $N_{Bin}\geq 100$; in fact, it can decline somewhat faster than for the QS approach. Clearly, for the *Gaussian* (blue) and *Exponential* (red) densities, the BC constant bin-width approximation can provide better entropy estimates with fewer bins than the QS variable bin-width approach. However, for the skewed *Log-Normal* density (orange) the behavior of the BC approximation is more complicated, whereas the QS approach shows an exponential rate of improvement with increasing number of bins for all three density types. This suggests that the variable bin-width QS approximation may provide a more consistent approach for more complex distributional forms (see Section 7).

Further, Figure 7 shows the results of a “*naïve*” implementation of BC where the value for $N_{Bin}$ is varied as a fractional percentage of sample size $N_{S}$. As with QS, we specify the support interval by setting $x_{min}=\min\left\{S\right\}$ and $x_{max}=\max\left\{S\right\}$, but here the support interval is divided into equal-width bins so that $X_{BIN}=\left\{X_{0},X_{1},X_{2},\ldots ,X_{N_{Bin}}\right\}$ represents the locations of the edges of the bins (where $X_{0}=x_{min}$ and $X_{N_{Bin}}=x_{max}$), and therefore $\Delta =\frac{x_{max}-x_{min}}{N_{Bin}}$. We then assume that $p\left(x\right)\approx \hat{p}\left(x|X_{BIN}\right)=\frac{n_{j}}{N_{S}}$ for $X_{j-1}\leq X<X_{j}$ where $n_{j}$ is the number of samples falling in the bin defined by $X_{j-1}\leq X<X_{j}$. Finally, we compute the BC estimate of entropy as ${\hat{H}}_{\hat{p}}\left(X|X_{BIN}\right)=-\overset{N_{Bin}}{\underset{j=1}{\sum }}ln\left(\frac{n_{j}}{N_{S}}\right)\cdot \frac{n_{j}}{N_{S}}+ln\left(\Delta \right)$, and follow the convention that $0\cdot ln\left(0\right)=0$ to handle bins where the number of samples $n_{j}=0$. Finally, we obtain estimates for 500 different sample sets drawn from the parent density and average the results. Results are shown for different sample sizes $N_{S}=$**{**100, 200, 500, 1000, 2000 and 5000}. The yellow marker symbols indicate where each curve crosses the zero-bias line; clearly $N_{Bin}$ is *not* a constant fraction of $N_{S}$, and for any given sample size the ratio of $\frac{N_{Bin}}{N_{S}}$ changes with form of the pdf.

To more clearly compare these BC results with the results shown in Figure 4 for the QS approach, Figure 8 shows a plot indicating the sampling variability distribution of the optimal number of bins (i.e., the value of $N_{Bin}$ for which the expected entropy estimation error is zero) as a function of sample size for the *Gaussian*, *Exponential* and *Log-Normal* densities. We see clearly that, in contrast to QS, the “*expected optimal*” number of bins to achieve zero bias is neither a constant fraction of the sample size or independent of the pdf shape, but instead declines as the sample size increases, and is different for different pdf shapes. Further, the sampling variability associated with the optimal fractional number of bins can be quite large, and is highly skewed at smaller sample sizes. This is in contrast with QS where the optimal fractional number of bins is approximately constant at α≈25–35% for different sample sizes and pdf shapes.

## 7. Testing on Multi-Modal PDF Forms

While the types of pdf forms tested in this paper are far from exhaustive, they represent differently shapes and degrees of skewness, including infinite support on both sides (*Gaussian*), and infinite support on only one side (*Exponential* and *Log-Normal*). However, all three forms are “*unimodal*”, and so we conducted an additional test for a multimodal distributional form.

Figure 9 shows results for a *Bimodal* pdf (Figure 9a) constructed using a mixture of two *Gaussians* $N\left(1,5\right)$ and $N\left(5,1\right)$. Since its theoretical entropy value is unknown, we used the piecewise constant approximation method with true (known) quantile positions to compute its entropy by progressively increasing the number of quantiles $N_{Z}$ until the estimate converged to within three decimal places (Figure 9b) to the value $H_{p}\left(X\right)\approx 2.265$ for $N_{Z}>2000$. Figure 9c,d show that QS estimation bias declines exponentially with fractional number of quantiles and crosses zero at α≈25% to 35%, in a manner similar to the *Unimodal* pdfs tested previously (Figure 4). Figure 9c shows the results for $N_{S}=5000$ samples, along with the distribution due to sampling variability (500 repetitions), showing that the IQR falls within ±1% of the correct value and the whiskers (±2 sigma) fall within ±3%. Figure 9d shows the expected bias (estimated by averaging over 500 repetitions) for varying sample size $N_{S}$; for smaller sample sizes, the optimal value for α is closer to 20%, while for $N_{S}\geq 200$ the value of α≈25% to 35% seems to work quite well, while being relatively insensitive to the choice of value within this range.

Interestingly, comparing Figure 9d (*Bimodal Gaussian Mixture*) with Figure 4a (*Unimodal Gaussian*), we see that QS actually converges more rapidly for the *Bimodal* density. One possible explanation is that the *Bimodal* density is in some sense “*closer*” in shape to a *Uniform* density, for which the piecewise constant representation is a better approximation.

## 8. Experimental Comparison with the Kernel Density Method

Figure 10 and Figure 11 show performance of the KD method on the same four distributions (*Gaussian, Exponential, Log-Normal* and *Bimodal*). We used the KD method developed by [21] and the code provided at webee.technion.ac.il/~yoav/research/blind-separation.html (accessed on 13 May 2021), which provides an efficient implementation of the non-parametric Parzen-window density estimator [4,22]. To align well with the known (exponential-type) shapes of the four example distributions, we used a *Gaussian* kernel, so that the corresponding hyper-parameter to be tuned/selected is the standard deviation $\sigma_{K}$ of each kernel. Consistent with our implementation of QS and BC, the code specifies the support interval by setting $x_{min}=\min\left\{S\right\}$ and $x_{max}=\max\left\{S\right\}$. As discussed below, the results clearly reinforce the findings of [17] that KD performance can depend strongly on distribution shape and sample size.

According to Parzen-window theory, the appropriate choice for the kernel standard deviation $\sigma_{K}$ is a function of the sample size $N_{S}$ such that $\sigma_{K}=K/\sqrt{N_{S}}$, where K is some unknown constant; this makes sense because for smaller sample sizes the data points are spaced further apart and we require more smoothing (larger $\sigma_{K}$), while for larger sample sizes the data points are more closely spaced and we require less smoothing (smaller $\sigma_{K}$). Accordingly, Figure 10 shows the expected percentage estimation error (bias) on the *y*-axis plotted as a function of $K=\sigma_{K}\cdot \sqrt{N_{S}}$ for different sample sizes (averaged over 500 repetitions) for each of the four distributions. The yellow marker symbols indicate where each curve crosses the zero-bias line. Clearly, in practice, the optimal value for the KD hyper-parameter K is not constant and varies as a function of both sample size and distributional form.

This variation is illustrated clearly by Figure 11 where the optimal K is plotted as a function of sample size $N_{S}$ for each of the four distributions. So, as with the BC (but in contrast with the QS), practical implementation of KD requires both selection of an appropriate kernel type *and* tuning to determine the optimal value of the kernel hyper-parameter (either K or $\sigma_{K}$); this can be done by optimizing $\sigma_{K}$ to maximize the *Likelihood* of the data. Note that for smaller sample sizes the entropy estimate can be very sensitive to the choice of this hyper-parameter, as evidenced by the steeper slopes of the curves in Figure 10. In contrast, with QS no kernel-type needs to be selected and no hyper-parameter tuning seems to be necessary, regardless of distribution shape and sample size, and sensitivity of the entropy estimate to precise choice of the number of quantiles $N_{Z}$ is relatively small even for smaller sample sizes (see Figure 4).

## 9. Discussion and Conclusions

The QS approach provides a relatively simple method for obtaining accurate estimates of entropy from data samples, along with an idea of the estimation-uncertainty associated with sampling variability. It appears to have an advantage over BC and KD since the most important hyper-parameter to be specified, the number of quantiles $N_{Z}$, does not need to be tuned and can apparently be set to a fixed fraction (~25–35%) of the sample size, regardless of pdf shape or sample size. In contrast, for BC the optimal number of bins $N_{Bin}$ varies with pdf shape and sample size and, since the underlying pdf shape is usually not known beforehand, it can be difficult to come up with a general rule for how to accurately specify this value. Similarly, for KD, without prior knowledge of the underlying pdf shape (and especially when the pdf may be non-smooth) it can be difficult to know what kernel-type and hyper-parameter settings to use.

Besides being simpler to apply, the QS approach appears to provide a more accurate estimate of the underlying data generating pdf than either BC or KD, particularly for smaller sample sizes. This is illustrated clearly by Figure 12 where the expected percent entropy estimation error is plotted as a function of sample size $N_{S}$ for each of the three methods. For QS, the fractional number of bins was fixed at α=25% regardless of pdf form or sample size; in other words, no hyperparameter tuning was performed. For each of the other methods, the corresponding hyperparameter (kernel standard deviation $\sigma_{K}$ for KD, and bin width **∆** for BC) was optimized for each random sample, by finding the value that maximizes the *Likelihood* of the sample. As can clearly be seen, the QS-based estimates remain relatively unbiased even for samples as small as 100 data points, whereas the KD- and BC-based estimates tend to get progressively worse (negatively biased) as sample sizes are decreased. Overall, QS is both easier to apply (no hyper-parameter tuning required) and likely to be more accurate than BC or KD when applied to data from an unknown distributional form, particularly since the the piecewise linear interpolation between CDF points makes it applicable to pdfs of any arbitrary shape, including those with sharp discontinuities in slope and/or magnitude. A follow-up study investigating the accuracy of these methods when faced with data drawn from complex, arbitrarily shaped, pdfs is currently in progress and will be reported in due course.

The fact that QS differs from BC in one very important way may help to explain the properties noted above. Whereas in BC we choose the “*bin*” size and locations, and then compute the “*probability mass*” estimates for each bin from the data, in QS we instead choose the “*probability mass*” size (by specifying the number of quantiles) and then compute the “*bin*” sizes and locations (to conform to the spacings between quantiles) from the data. In doing so, BC uses only the samples falling within a particular bin to compute each probability mass estimate, which value can (in principle) be highly sensitive to sampling variability unless the number of samples (in each bin) is sufficiently large. In contrast, QS uses a potentially large number of samples from the data to generate a smoothed (via subsampling and averaging) estimate of the position of each quantile. As shown in Figure 2, the estimation bias and uncertainty are small for most of the quantiles and may only be significant near the extreme tails of the density, and for smaller sample sizes. In principle, therefore, with its focus on estimating quantiles rather than probability masses, the QS method seems to provide a more efficient use of the information in the data, and thereby a more robust approximation of the shape of the pdf.

In this paper, we have used a simple, perhaps naïve, way of estimating the locations of the quantiles. Future work could investigate more sophisticated methods where the bias associated with extreme quantiles is accounted for and corrected. These include both Kernel and non-parametric methods. The simplest non-parametric methods are the empirical quantile estimator based on a single order statistic, or the extension based on two consecutive order statistics [23], for which the variance can be large. Quantile estimators based on L-statistics have been explored as a way to reduce estimation variance [24,25,26], and include Kernel quantile estimators [27,28,29,30,31]. However, performance of the latter can be very sensitive to the choice of bandwidth. More recently, quantile L-estimators intended to be efficient at small sample sizes for estimating quantiles in the tails of a distribution have also been proposed [32,33]. Finally, quantile estimators based on Bernstein polynomials [34,35] and importance sampling (see [36] and references therein) have been also investigated.

Note that the small-sample efficiency of the QS method may be affected by the fact that the entropy fractions associated with the extreme upper and lower end “*bins*” (quantile spacings) can be quite large when a small number of quantiles is used (see Figure 3). Our implementation of QS seems to successfully compensate for lack of exact knowledge of $z_{0}$ and $z_{N_{Z}}$ by using as empirical estimates the values $x_{min}$ and $x_{max}$ in the data sample. Intuitively, one would expect that wider “*bins*” could be used in regions where the slopes of the entropy fraction curves are flatter (e.g., the center of the *Gaussian* density), and narrower “*bins*” in the tails (more like the BC method). Taken to its logical conclusion, an ideal approach might be to use “*bin*” locations and widths such that the cumulative value of $-ln\left(p\right)\cdot p$ for any given pdf is subdivided into equal intervals (i.e., equal fractional entropy spacings) such that each bin then contributes approximately the same amount to the summation in Equation (9). To achieve this, the challenge is to estimate the edge locations of these “*bins*” (analogous to locations of the quantiles) from the sample data; we leave the possibility of such an approach for future investigation.

## Figures and Tables

**Figure 1 entropy-23-00740-f001:**
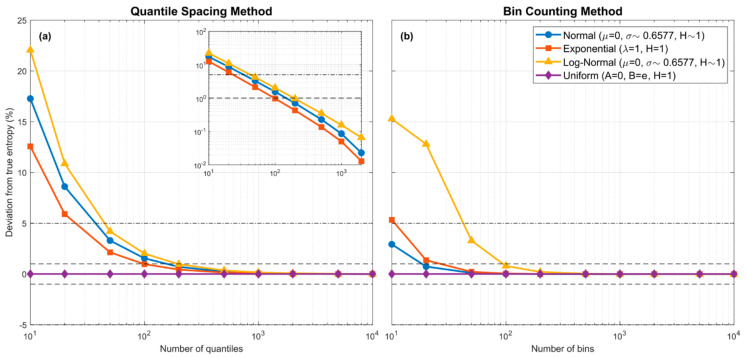
Plots showing how entropy estimation bias associated with the piecewise-constant approximation of various theoretical pdf forms varies with the number of quantiles ((**a**); QS method) or number of equal-width bins ((**b**); BC method) used in the approximation. The dashed horizontal lines indicate ± 1% and ± 5% bias error. No sampling is involved and the bias is due purely to the piecewise constant assumption. For QS, the locations of the quantiles are set to their theoretical values. To address the “*infinite support*” issue, $\left[x_{min},x_{max}\right]$ were set to be the locations where $P\left(z_{0}\right)=\varepsilon$ and $P\left(z_{N_{Z}}\right)=1-\varepsilon$ respectively, with $\varepsilon =10^{-5}$. In both cases, bias approaches zero as the number of piecewise-constant units is increased. For the QS method, the decline in bias is approximately linear in the log-log space (see inlay in the left subplot).

**Figure 2 entropy-23-00740-f002:**
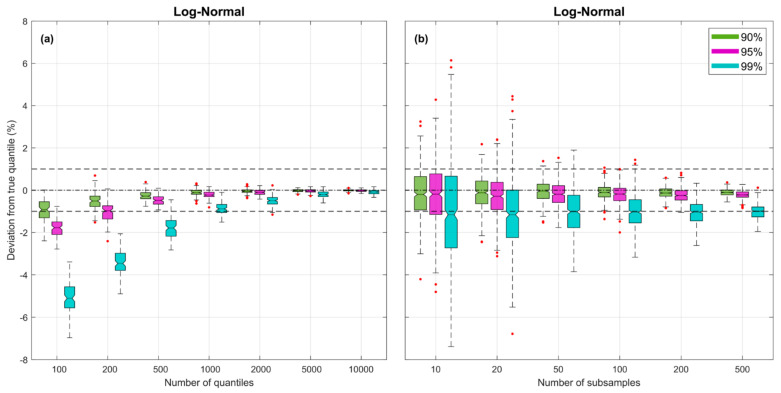
Plots showing bias and uncertainty associated with estimates of the quantiles derived from random samples, for the *Log-Normal* pdf. Uncertainty associated with random sampling variability is estimated by repeating each experiment 500 times. In both subplots, for each case, the box plots are shown side by side to improve legibility. (**a**) Subplot showing results varying $N_{Z}=\left[100,\;200,\;500,\;1000,\;2000,\;5000,\;10000\right]$ for fixed $N_{K}=500$. (**b**) Subplot showing results varying $N_{K}=\left[10,\;20,\;50,\;100,\;200,\;500\right]$ for fixed $N_{Z}=1000$.

**Figure 3 entropy-23-00740-f003:**
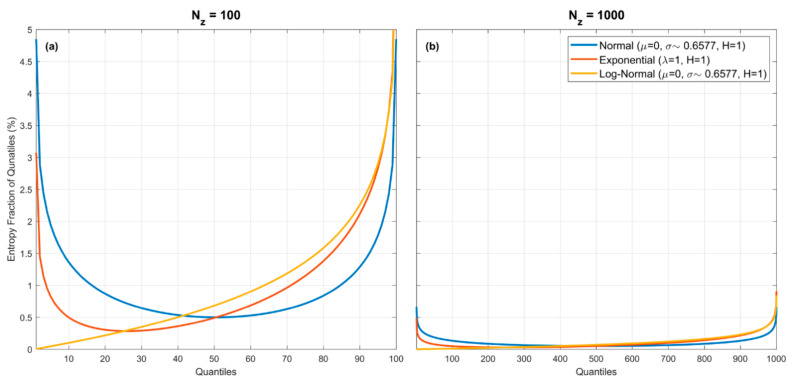
Plots showing percentage entropy fraction associated with each quantile spacing for the *Gaussian*, *Exponential* and *Log-Normal* pdfs, for $N_{Z}=100$ (**a**), and $N_{Z}=1000$ (**b**). For the *Uniform* pdf (not shown to avoid complicating the figures) the percentage entropy fraction associated with each quantile spacing is a horizontal line (at 1% in the left panel, and at 0.1% in the right panel). Note that the entropy fractions can be proportionally quite large or small at the extremes, depending on the form of the pdf. However, the overall entropy fraction associated with each quantile spacing diminishes with increasing $N_{Z}.$ For the examples shown, the maximum contributions associated with a quantile spacing are less than 6% for $N_{Z}=100$ (**a**), and become less than 1% for $N_{Z}=1000$ (**b**).

**Figure 4 entropy-23-00740-f004:**
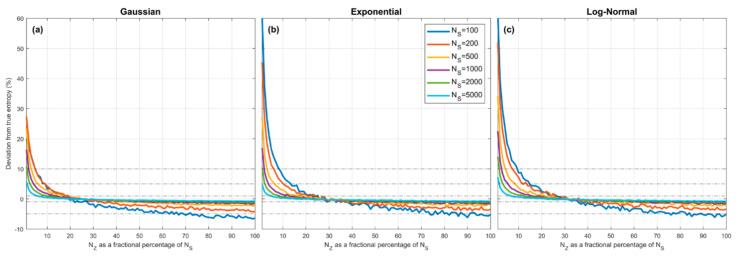
Plots showing expected percent error in the QS-based estimate of entropy derived from random samples, as a function of $\alpha =100*\left(N_{Z}/N_{S}\right)$, which expresses the number of quantiles $N_{Z}$ as a fractional percentage of the sample size $N_{S}$. Results are averaged over 500 trials obtained by drawing sample sets of size $N_{S}$ from the theoretical pdf, where $x_{min}$ and $x_{max}$ are set to be the smallest and largest data values in the particular sample. Results are shown for different sample sizes $N_{S}=\left[100,\;200,\;500,\;1000,\;2000,\;5000\right]$, for the *Gaussian* (**a**), *Exponential* (**b**) and *Log-Normal* (**c**) densities. In each case, when α is small the estimation bias is positive (overestimation) and can be greater than 10% for α<10%, and crosses zero to become negative (underestimation) when α>25–35%. The marginal cost of setting α too large is low compared to setting α too small. As $N_{S}$ increases, the bias diminishes. The optimal choice is α≈25–30% and is relatively insensitive to pdf shape or sample size.

**Figure 5 entropy-23-00740-f005:**
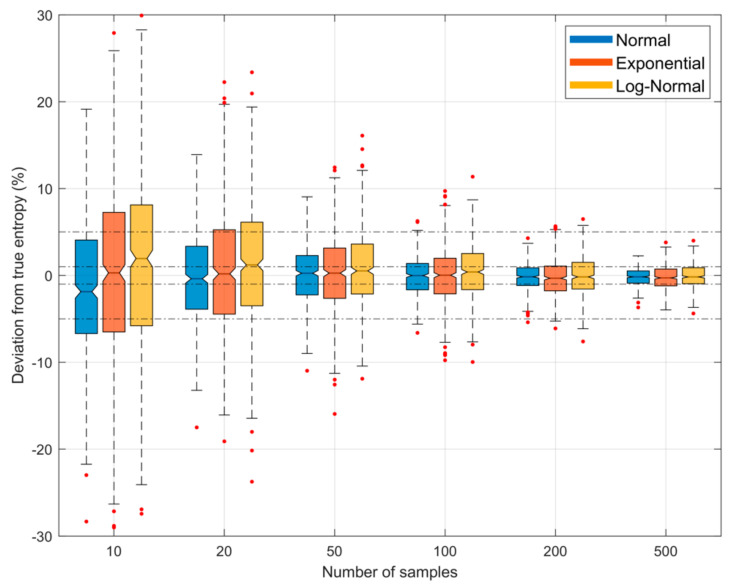
Plots showing bias and uncertainty in the QS-based estimate of entropy derived from random samples, as a function of sample size $N_{S}$, when the number of quantiles $N_{Z}$ is set to 25% of the sample size (α=0.25), and $x_{min}$ and $x_{max}$ are respectively set to be the smallest and largest data values in the particular sample. The uncertainty shown is due to random sampling variability, estimated by drawing 500 different samples from the parent density. Results are shown for the *Gaussian* (blue), *Exponential* (red) and *Log-Normal* (orange) densities; box plots are shown side by side to improve legibility. As sample size $N_{S}$ increases, the uncertainty diminishes.

**Figure 6 entropy-23-00740-f006:**
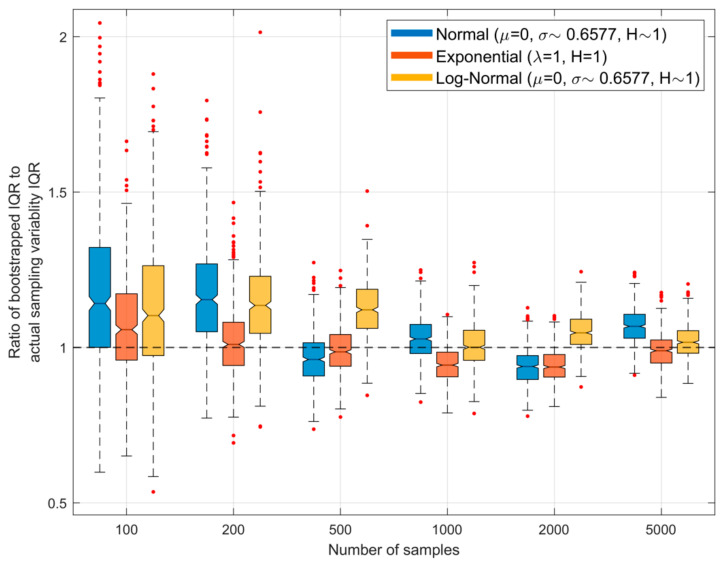
Plots showing, for different sample sizes and α=25%, the ratio of the interquartile range (IQR) of the QS-based estimate of entropy obtained using bootstrapping to that of the actual IQR arising due to random sampling variability. Here, each sample set drawn from the parent density is bootstrapped to obtain $N_{B}=500$ different estimates of the associated entropy, and the width of the resulting inter-quartile range is computed. The procedure is repeated for 500 different sample sets drawn from the parent population, and the graph shows the resulting variability as box-plots. The ideal result would be a ratio of 1.0.

**Figure 7 entropy-23-00740-f007:**
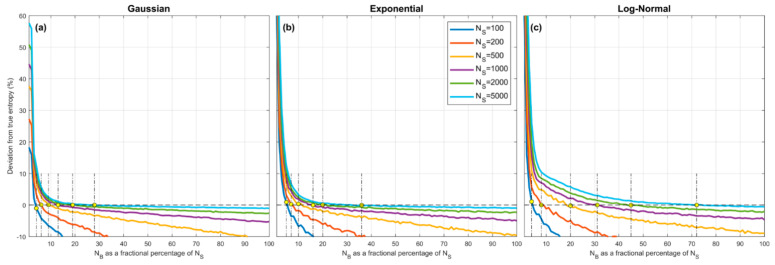
Plots showing how expected percentage error in the BC-based estimate of Entropy derived from random samples, varies as a function of the number of bins $N_{Bin}$ for the (**a**) *Gaussian*, (**b**), *Exponential*, and (**c**) *Log-Normal* densities. Results are averaged over 500 trials obtained by drawing sample sets of size $N_{S}$ from the theoretical pdf, where $x_{min}$ and $x_{max}$ are set to be the smallest and largest data values in the particular sample. Results are shown for different sample sizes $N_{S}=\left[100,\;200,\;500,\;1000,\;2000,\;5000\right]$. When the number of bins is small the estimation bias is positive (overestimation) but rapidly declines to cross zero and become negative (underestimation) as the number of bins is increased. In general, the overall ranges of overestimation and underestimation bias are larger than for the QS method (see Figure 4).

**Figure 8 entropy-23-00740-f008:**
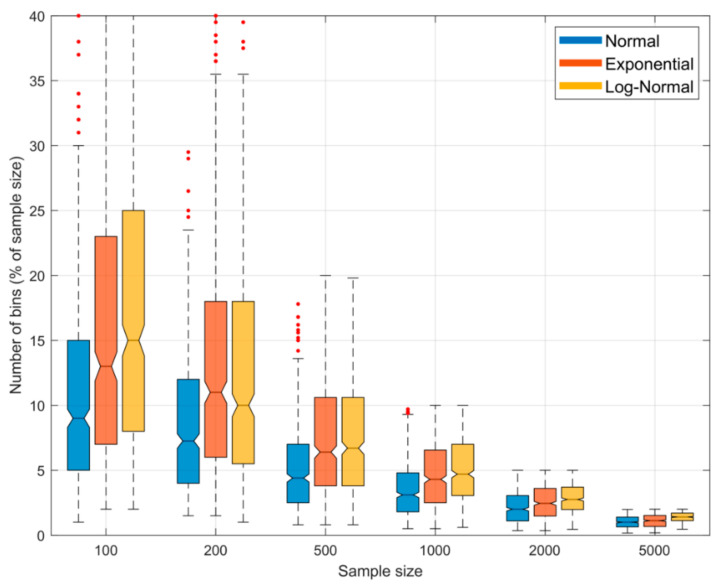
Boxplots showing the sampling variability distribution of optimal fractional number of bins (as a percentage of sample size) to achieve zero bias, when using the BC method for estimating entropy from random samples. Results are shown for the *Gaussian* (blue), *Exponential* (red) and *Log-Normal* (orange) densities. The uncertainty estimates are computed by drawing 500 different sample data sets of a given size from the parent distribution. Note that the expected optimal fractional number of bins varies with shape of the pdf, and is not constant but declines as the sample size increases. This is in contrast with the QS method where the optimal fractional number of bins is constant at ~25% for different sample sizes and pdf shapes. Further, the variability in optimal fractional number of bins can be large and highly skewed at smaller sample sizes.

**Figure 9 entropy-23-00740-f009:**
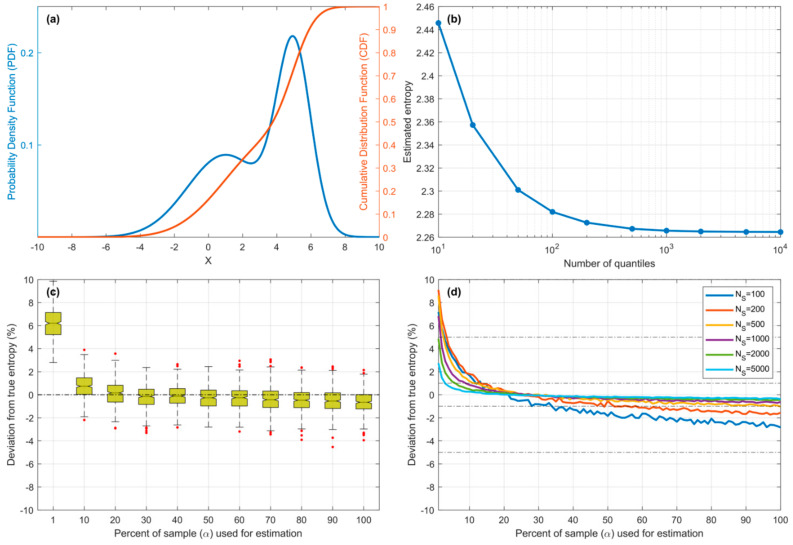
Plots showing results for the *Bimodal* pdf. (**a**) Pdf and Cdf for the *Gaussian Mixture* model. (**b**) Showing convergence of entropy computed using piecewise constant approximation as the number of quantiles $N_{Z}$ is increased. (**c**) Bias and sampling variability of the QS-based estimate of entropy plotted against $N_{Z}$ as a percentage of sample size. (**d**) Expected bias of QS-based estimate of entropy plotted against $N_{Z}$ as a percentage of sample size, for different sample sizes $N_{S}=\left[100,\;200,\;500,\;1000,\;2000,\;5000\right]$.

**Figure 10 entropy-23-00740-f010:**
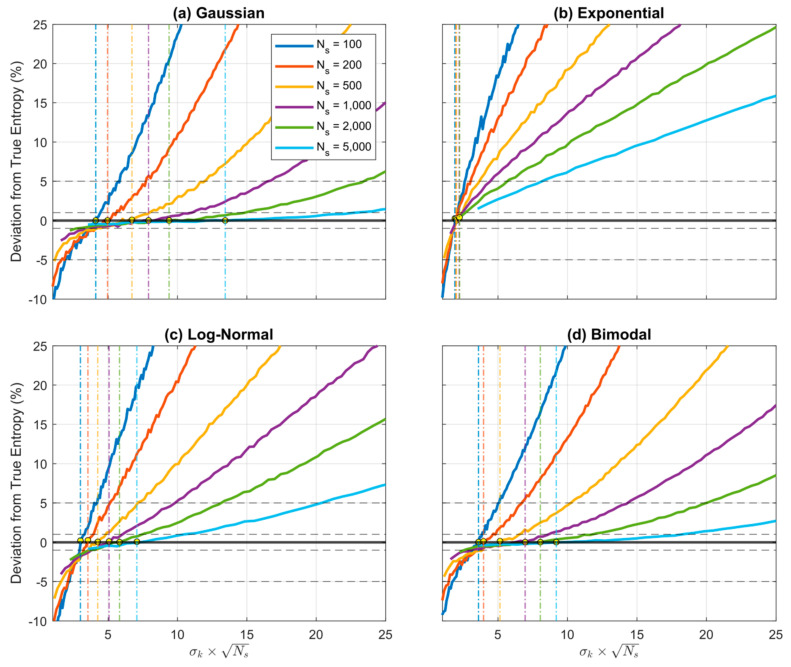
Plot showing how expected percentage error in the KD-based estimate of Entropy derived from random samples, varies as a function of $K=\sigma_{k}\cdot \sqrt{N_{S}}$ when using a *Gaussian* kernel. Results are averaged over 500 trials obtained by drawing sample sets of size $N_{S}$ from the theoretical pdf, where $x_{min}$ and $x_{max}$ are set to be the smallest and largest data values in the particular sample. Results are shown for different sample sizes $N_{S}=\left[100,\;200,\;500,\;1000,\;2000,\;5000\right]$, for the (**a**) *Gaussian*, (**b**), *Exponential*, (**c**) *Log-Normal*, and (**d**) *Bimodal* densities. When the kernel standard deviation $\sigma_{k}$ (and hence K) is small the estimation bias is negative (underestimation) but rapidly increases to cross zero and become positive (overestimation) as the kernel standard deviation is increased. The location of the crossing point (corresponding to optimal value for K (and hence $\sigma_{k}$) varies with sample size and shape of the pdf.

**Figure 11 entropy-23-00740-f011:**
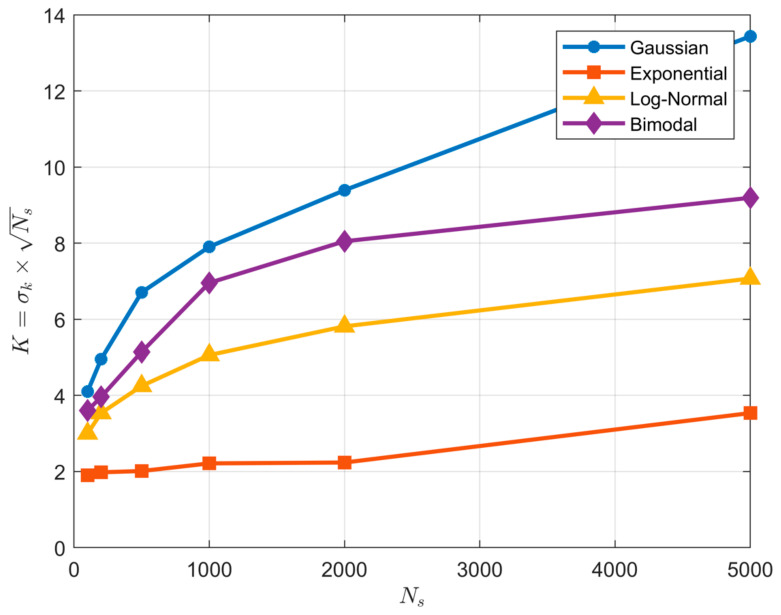
Plot showing how the optimal value of the KD hyper-parameter $K=\sigma_{k}\cdot \sqrt{N_{S}}$ varies as a function of sample size $N_{S}$ and pdf type when using a *Gaussian* kernel. In disagreement with Parzen-window theory, the optimal value for K does not remain approximately constant as the sample size $N_{S}$ is varied. Further, the value of K varies significantly with shape of the underlying pdf.

**Figure 12 entropy-23-00740-f012:**
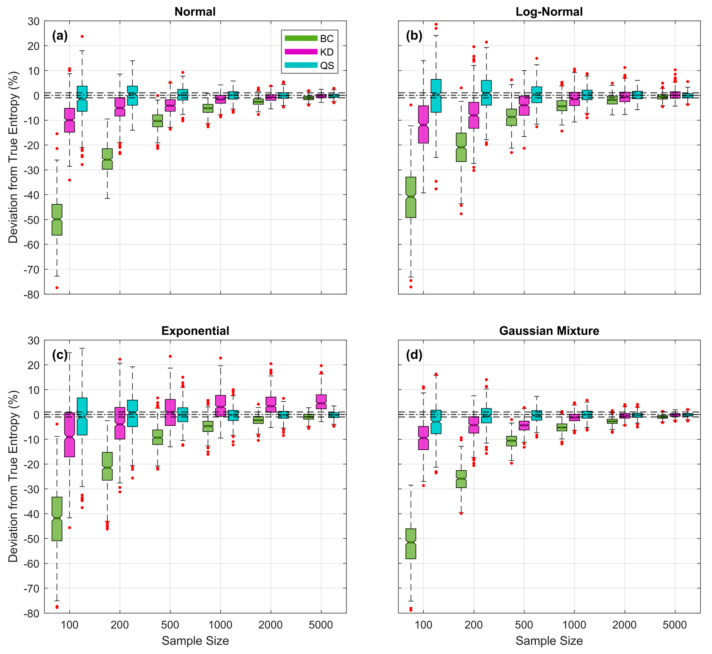
Plots showing expected percent error in the QS- (blue), KD- (purple) and BC-based (green) estimates of entropy derived from random samples, as a function of sample size $N_{S}$ for the (**a**) *Gaussian*, (**b**), *Log-Normal*, (**c**) *Exponential*, and (**d**) *Bimodal* densities; box plots are shown side by side to improve legibility. Results are averaged over 500 trials obtained by drawing sample sets of size $N_{S}$ from the theoretical pdf, where $x_{min}$ and $x_{max}$ are set to be the smallest and largest data values in the particular sample. For QS, the fractional number of bins was fixed at α=25% regardless of pdf form or sample size. For KD and BC, the corresponding hyperparameter (kernel standard deviation $\sigma_{K}$ and bin width ∆ respectively) was optimized for each random sample by finding the value that maximizes the *Likelihood* of the sample. Results show clearly that QS-based estimates are relatively unbiased, even for small sample sizes, whereas KD- and BC-based estimates can have significant negative bias when sample sizes are small.

## Data Availability

No new data were created or analyzed in this study. Data sharing is not applicable to this article.

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
