# Peer review of "Computing Accurate Probabilistic Estimates of One-D Entropy from Equiprobable Random Samples"

_entropy, 2021, doi:10.3390/e23060740_

Round 1

Reviewer 1 Report

The idea presented in the paper is interesting but

  • some references in the text are missing in the bibliography eg. Pepelyshev, Andrey, E. Rafajłowicz, and A. Steland. "Estimation of the quantile function using Bernstein–Durrmeyer polynomials." Journal of Nonparametric Statistics 26.1 (2014): 1-20.
  • the idea presented is very similar to a one-dimensional version of Loftsgarden and Quesenberry estimator " A nonparametric estimate of a multivariate density function. The Annals of Mathematical Statistics36(3), 1049-1051."
  • no theoretical results are presented but with know distributions, they can possibly be obtained
  • numerical simulations, in my opinion, to weakly support the presented method as better than calculating a simple histogram
  • no example regarding real data are presented,

to sum up, the paper requires major work.

Author Response

Reviewer #1:

Comment: The idea presented in the paper is interesting but

  • some references in the text are missing in the bibliography eg. Pepelyshev, Andrey, E. Rafajłowicz, and A. Steland. "Estimation of the quantile function using Bernstein–Durrmeyer polynomials." Journal of Nonparametric Statistics1 (2014): 1-20.
  • the idea presented is very similar to a one-dimensional version of Loftsgarden and Quesenberry estimator " A nonparametric estimate of a multivariate density function. The Annals of Mathematical Statistics, 36(3), 1049-1051."
  • no theoretical results are presented but with know distributions, they can possibly be obtained by numerical simulations, in my opinion, to weakly support the presented method as better than calculating a simple histogram
  • no example regarding real data are presented,

to sum up, the paper requires major work.

Response: Thanks for these suggestions.

  • Thanks for pointing out the missing reference. Apologies for this omission that has been rectified in the revised manuscript.
  • Thanks also for pointing us to the theoretical contribution of Loftsgarden and Quesenberry (1965) which deals with the non-parametric estimation of a multi-variate density function and proposes a density estimation method that does appear to be broadly theoretically related to our QS approach, as (of course) is any available other density estimation methodology. In their comments L&Q remark that “based on empirical work, a value of  near  appears to give good results” where  is defined as some “non-decreasing sequence of positive integers” that must be chosen by the user and  is the number of data points. We note that they actually focus on establishing an estimate of the “density”  around a point . Further, we are unable to find an algorithmic implementation or any suggestions for how  should be selected, other than the constraints that  and . 

In our approach we have proposed using the positions of the quantiles empirically estimated from the data, rather than some “non-decreasing sequence of positive integers” (while L&Q indicate that this positive integer condition can be generalized to more general  with “minor” difficulty, they do not explain further how to do so).  Further, we are not essentially attempting to directly estimate the “density” near any point  (actually, the whole point is to avoid having to do so), and instead proceed by assuming that the CDF can be approximated to arbitrary accuracy by assuming piecewise-linear variation between quantiles (equivalent to assuming piecewise constant density between quantiles).  The essential challenge is estimating the locations of the quantiles and we propose an approach that seems to work well in practice.

  • Regarding theoretical results, it is easy (indeed trivial) to show that the KL Divergence between the true and approximated pdf’s approaches zero as the number of quantiles is increased towards infinity, regardless of distributional form, and this is illustrated already by Figure 1 for both the QS and BC methods. Further demonstration seems unnecessary. Regarding numerical simulations we have indeed already provided extensive simulations to demonstrate the performance of the method on some theoretically known distributions of various kinds.  Further, we have clearly shown the effects of random sampling variability and how to compute uncertainty estimates, via bootstrapping, to account for such the unavoidable uncertainty arising due to such variability.
  • Regarding providing examples with real data (which would unavoidably be of finite sample sizes), such examples would be subject to the condition that the true underlying distributional form, and hence its entropy will unavoidably be unknown, and such experiments would therefore provide no convincing evidence to support the viability of the QS method (or indeed any other entropy estimation method). While, we certainly intend to use the QS method in real applications (which is why we developed it in the first place), we don’t see how that would advance the objective of this paper, which is to demonstrate that the QS approach functions as theoretically and conceptually expected and gives accurate results under conditions where the correct answer can be known.  In cases where the correct answer cannot be known, running real data experiments would not demonstrate the viability of the method.

Proposed Revision: In light of the reviewer’s comments, we propose adding the following changes to the manuscript to point out the connection to the theoretical contribution of Loftsgarden and Quesenberry (1965).  Further, we have checked carefully to ensure that all references are properly cited.

[New Para 8] We focus here on the case of a one-dimensional continuous random variable for which the mathematical form of  is unknown. Silverman (1986) provides a summary of methods for the estimation of pdfs from data, while Beirlant et al (1997) provides an overview of methods for estimating the differential entropy of a continuous random variable. The three most widely used “non-parametric” methods for estimating differential entropy by pdf approximation are the: (a) Bin-Counting (BC) or piece-wise constant frequency histogram method, (b) Kernel Density (KD) method, and (c) Average Shifted Histogram (ASH) method. Scott (2008) points out that these can all be asymptotically viewed as ‘‘Kernel’’ methods, where the bins in the BC and ASH approaches are understood as treating the data points falling in each bin as being from a locally uniform distribution. For a theoretical discussion of the basis for non-parameteric estimation of a density function see also Loftsgarden and Quesenberry (1965).

Reviewer 2 Report

This paper proposes a variable bin size for estimating differential entropy of continuous random variables.

I have no issues with the proposed approach, but I do feel like some comparisons are missing.  Kernel density methods are mentioned in the Introduction, and there are some good ones (kNN, Parzen window estimates, even neural network-based approaches that are in fact parametric) that one could use to get parametric or nonparametric estimates of the pdf and then from that get the differential entropy.  Also, there is a well-known kNN estimator of differential entropy that should be compared to.  If these comparisons are done, I have no issue recommending acceptance.

Author Response

Reviewer #2:

Comment: This paper proposes a variable bin size for estimating differential entropy of continuous random variables.

I have no issues with the proposed approach, but I do feel like some comparisons are missing.  Kernel density methods are mentioned in the Introduction, and there are some good ones (kNN, Parzen window estimates, even neural network-based approaches that are in fact parametric) that one could use to get parametric or nonparametric estimates of the pdf and then from that get the differential entropy.  Also, there is a well-known kNN estimator of differential entropy that should be compared to.  If these comparisons are done, I have no issue recommending acceptance.

Response: Thanks for these suggestions. Indeed, as the reviewer points out, several approaches exist to estimate differential entropy, either with or without the intermediate step of estimating a pdf. The most popular approaches, Bin-counting (BC), kernel density methods (KD) and average shifted histogram (ASH) we present and briefly discuss in section 2 of the paper. However, for an in-depth comparison of the proposed quantile-spacing (QS) method -- in terms of properties and performance in test applications -- we have chosen to only present for BC.

The reasons for this are several:

  • The first is that BC has some obvious similarities with QS, but also some important and interesting differences, and we consider it important to bring these issues to the attention of the reader (see discussion in section 5).
  • Secondly, BC is arguably the most straightforward (easy to understand and implement) and popular of the alternative approaches.

For these reasons, together with our goal of providing (by this paper) a comprehensive yet concise presentation of the QS method, we have decided to provide an in-depth comparison to only one method, and the most obvious and relevant choice for this method happens to be BC.

Like the referee, we initially also considered including KD as an additional benchmark, due to its popularity and generality. After some consideration, we decided not to do so for several reasons. 

  • First, importantly, is to maintain brevity of the manuscript.
  • Second, and perhaps more important, is that KD suffers from some important limitations that do not apply to the QS approach. In particular, an important characteristic of QS is that no assumptions whatsoever are required to be made regarding the shape of the underlying data generating process (the pdf) under investigation, and hence it is applicable to any kind of density, uni-modal, bi-modal or other. In fact, the only relevant assumption is that the cumulative density function (CDF) can be approximated arbitrarily closely as being piecewise-linear between quantiles, such that the KL Divergence between the assumed and true pdf’s rapidly approaches zero uniformly as the number of quantiles is increased towards infinity (see Figure 1).
  • Third, in Darscheid et al. (2018), we have previously investigated the suitability of several methods to estimate discrete distributions from limited samples, including BC and KD. We did so for a range of candidate distributions (see Fig. 2 from Darscheid et al. (2018) included below) and measured performance using the Kullback-Leibler divergence (see Fig. 3 from Darscheid et al. (2018) included below). Compared to other methods, the performance of KD proved to be strongly dependent on the distribution shape which, unfortunately, limits the generality of its application.
  • Finally, parameteric methods such as KD require some kind of hyper-parameter optimization before the results can be usable. Of course, as has been pointed out in the literature, BC can also be viewed as a KD method using uniform density kernels, with the consequence being that an “optimal” bin-width has to be selected for each application (see e.g., Scott 1978, 2004, 2008 etc). In our study (Figure 7), we avoid doing this by reporting results for a wide range of BC bin-widths and show clearly that the results are very sensitive to such. In contrast, the QS approach requires no such hyper-parameter optimization; one simply selects the number of quantiles to be used in computing the pdf approximation to be 25-30% of the sample size, and our results (Figure 4 and 9d) show clearly that this choice results in entropy estimates within 1% of the true value for density functions of varying shapes, including skewed and bi-modal.

One of our major conclusions is that the QS approach, implemented as suggested in our paper, avoids the need for either subjective decisions (shape of kernel, etc) or optimization. In this sense the QS approach is truly a “plug-and-play” methodology – apply the algorithm to the data and obtain a high-quality estimate of the entropy (along with uncertainty estimates) without the need for situational dependent assumptions or any kind of optimization/fitting/tuning. 

Overall, for the reasons given above, this is why we have used BC, and BC only as the basis for our comparison.

Proposed Revision: Given that the reviewer raised this issue, we therefore think that it is important to address it in the manuscript, as many of the intended readers may have similar thoughts. We have propose to adding the following sentence, addressing this point, at the end of section 2:

[New Para 12] In summary, while BC, KD and ASH can be all be used to obtain accurate estimates of entropy for pdfs, hyper-parameter tuning is unavoidably required to ensure that good results are obtained. In the next section, we propose an alternative method to approximate  that does not require counting the numbers of samples in “bins”, and is instead based on estimating the quantile positions of . We compare the properties and performance of the method only with BC (rather than with KD or ASH) for a range of distributions because i) BC is arguably the most straightforward and popular method, ii) BC shares important similarities with the proposed approach, but also shows distinct differences worthy of discussion, and iii) we seek good performance for a wide range of distributions, whereas KD performance can strongly depend on distribution shape (Darscheid et al., 2018).

New References Cited

Darscheid, P., A. Guthke, and U. Ehret (2018), A Maximum-Entropy Method to Estimate Discrete Distributions from Samples Ensuring Nonzero Probabilities, Entropy, 20(8), 601.

Below à Figures from Darscheid et al (2018) mentioned in our response above:

Round 2

Reviewer 1 Report

The corrections in the paper are sufficient

Author Response

Reviewer #2:

Comment: I appreciate the response to my request for comparison to other methods, but I have to disagree with the assessment that the hyperparameter tuning is cumbersome and difficult to do.  Pseudolikelihood methods for Parzen window estimates are actually incredibly easy to implement and quite effective.  I must insist on comparison to some of these other methods that have hyperparameters.  I am fine with a briefly-described result that says that this new method is worse due to the fact that it doesn't have as much to tune, but that should be known.  I would actually say that you have a hyperparameter, which is how many quantiles are used, and so are on par with, say, Parzen window estimates for difficulty.

Response: Thanks for the suggestions, and the push to perform the additional comparison, which proved to be very interesting.

The revised paper now includes a comparison with the Parzen-window Kernel-Density (KD) method using a Gaussian-type kernel, which seems most appropriate for the exponential type densities (Gaussian, Exponential, Log-Normal and Bimodal Gaussian Mixture) used in our experiments. 

In particular we used the KD entropy estimation method developed by Schwartz et al (2005) and the code provided at webee.technion.ac.il/~yoav/research/blind-separation.html, which provides an efficient implementation of the non-parametric Parzen-window density estimator, to obtain estimates of entropy from a data sample.

Schwartz S, Zibulevsky M and YY Schechner (2005), Fast kernel entropy estimation and optimization, Signal Processing 85, pp. 1045–1058.

We ran the KD method on all four of the test pdfs (Gaussian, Exponential, Log-Normal and Bimodal Gaussian), using the exact same experimental design as for QS and BC, where (for each sample size) 500 replications were performed for each pdf to account for sampling variability. Note that the pdf support was computed based on the minimum and maximum value in the sample, and the computed entropy estimates were averaged to give expected values. This ensures that we provide a direct and meaningful comparison to the QS and BC results reported in the paper.

Specifically, we computed the percent expected error (bias) in the KD entropy estimate for different values of the Gaussian kernel hyper-parameter (the standard deviation  of the kernels), and plotted the value of the bias against the value , where  is the sample size, for different sample sizes from 100 to 5000.  According to Parzen-window theory, , meaning that the value  can be expected to be some constant value for any given pdf and should therefore not depend (significantly) on the sample size . 

The new Figure 10 shows the percent expected error (bias) in the entropy estimate plotted against the product  for all four of the test pdfs. As can be seen, for any given pdf, the value of  (and hence ) at which zero estimation bias is achieved is, in practice, not a constant value for different sample sizes . Further, the relationship between optimal  (and hence ) and the sample size  is quite different for the different pdfs, as illustrated in the new Figure 11.

The new Figure 10 also shows that for smaller sample sizes (e.g., ) the entropy estimate can be quite sensitive to the choice of the KD hyper-parameter, as evidenced by the steeper slopes of the curves.

So, in general, when implementing the KD approach, the user must both select the appropriate kernel-type, and tune the kernel width hyperparameter for each data set and sample size to obtain accurate results (defined in our paper as being within  of the true value).  In contrast, our results show that with QS no kernel-type needs to be selected and no hyper-parameter tuning seems to be necessary, regardless of distribution shape and sample size. Further, in comparison with KD, the relative sensitivity of the QS estimate of entropy to the precise choice of the number of quantiles  is relatively low even for sample sizes as small as  (see Figure 4). Finally, although not demonstrated explicitly in this paper, the QS method is theoretically applicable to pdfs of arbitrary shape, including those with discontinuous slope and/or magnitude (we plan to explore this desirable property of QS in future work).

Accordingly, it would not be appropriate for us to conclude that the QS method provides “worse” results than KD (as implied above by the reviewer).  In fact, the presented results indicate that it can provide results of comparable accuracy (within  of the true value) without the need for any kernel-type selection or hyperparameter tuning.  Further, our results indicate that the QS hyperparameter can be set at between 25-35% to obtain an estimate within  of the true value, regardless of pdf type, thereby obviating the need for any additional data sample specific tuning steps.

Proposed Revision: In light of these findings, we propose making the following changes to the manuscript to include the new (reviewer requested) comparison, and to illustrate the relative properties of the QS, BC and KD methods (new text shown in red).

  • The Abstract has been modified to read as follows:

[New Abstract] We develop a simple Quantile Spacing (QS) method for accurate probabilistic estimation of one-dimensional entropy from equiprobable random samples, and compare it with the popular Bin-Counting (BC) and Kernel Density (KD) methods. In contrast to BC, which uses equal-width bins with varying probability mass, the QS method uses estimates of the quantiles that divide the support of the data generating probability density function (pdf) into equal-probability-mass intervals. And, whereas BC and KD each require optimal tuning of a hyper-parameter whose value varies with sample size and shape of the pdf, QS only requires specification of the number of quantiles to be used.

Results indicate, for the class of distributions tested, that the optimal number of quantiles is a fixed fraction of the sample size (empirically determined to be ~0.25-0.35), and that this value is relatively insensitive to distributional form or sample size. This provides a clear advantage over BC and KD since hyper-parameter tuning is not required. Further, unlike KD, there is no need to select an appropriate kernel-type, and so QS is applicable to pdfs of arbitrary shape, including those with discontinuous slope and/or magnitude.

Bootstrapping is used to approximate the sampling variability distribution of the resulting entropy estimate, and is shown to accurately reflect the true uncertainty. For the four distributional forms studied (Gaussian, Log-Normal, Exponential and Bimodal Gaussian Mixture), expected estimation bias is less than 1% and uncertainty is relatively low even for very small sample sizes. We speculate that estimating quantile locations, rather than bin-probabilities, results in more efficient use of the information in the data to approximate the underlying shape of an unknown data generating pdf.

  • Paragraph 12 has been modified as follows:

[New Para 12] In summary, while BC and KD can be used to obtain accurate estimates of entropy for pdfs of arbitrary form, hyper-parameter tuning is required to ensure that good results are obtained. In the next section, we propose an alternative method to approximate  that does not require counting the numbers of samples in “bins”, and is instead based on estimating the quantile positions of . We first compare the properties and performance of the method with BC (rather than with KD or ASH) for a range of distributions because i) BC is arguably the most straightforward and popular method, ii) BC shares important similarities with the proposed approach, but also shows distinct differences worthy of discussion, and iii) we seek good performance for a wide range of distributions. At the request of a reviewer, we further report performance of KD on the same distributions, reinforcing the findings of Darscheid et al., (2018) that KD performance can depend strongly on distribution shape and sample size.

  • We have added a new Section 8 entitled “Experimental Comparison with the KD Method”, which consists of three new paragraphs as follows:

[New Para 53] Figures 10 and 11 show performance of the KD method on the same four distributions (Gaussian, Exponential, Log-Normal and Bimodal). We used the KD method developed by Schwartz et al (2005) and the code provided at webee.technion.ac.il/~yoav/research/blind-separation.html, which provides an efficient implementation of the non-parametric Parzen-window density estimator (Silverman 1986, Viola 1995). To align well with the known (exponential-type) shapes of the four example distributions, we used a Gaussian kernel, so that the corresponding hyper-parameter to be tuned/selected is the standard deviation  of each kernel. Consistent with our implementation of QS and BC, the code specifies the support interval by setting  and . As discussed below, the results clearly reinforce the findings of Darscheid et al., (2018) that KD performance can depend strongly on distribution shape and sample size.

[New Para 54] According to Parzen-window theory, the appropriate choice for the kernel standard deviation  is a function of the sample size  such that , where  is some unknown constant; this makes sense because for smaller sample sizes the data points are spaced further apart and we require more smoothing (larger ), while for larger sample sizes the data points are more closely spaced and we require less smoothing (smaller ). Accordingly, Figure 10 shows the expected percentage estimation error (bias) on the y-axis plotted as a function of  for different sample sizes (averaged over 500 repetitions) for each of the four distributions. The yellow marker symbols indicate where each curve crosses the zero-bias line.  Clearly, in practice, the optimal value for the KD hyper-parameter  is not constant and varies as a function of both sample size and distributional form. 

[New Para 55] This variation is illustrated clearly by Figure 11 where the optimal  is plotted as a function of sample size  for each of the four distributions. So, as with the BC (but in contrast with the QS), practical implementation of KD requires both selection of an appropriate kernel type and tuning to determine the optimal value of the kernel hyper-parameter (either  or ); this can be done by optimizing  to maximize the Likelihood of the data. Note that for smaller sample sizes the entropy estimate can be very sensitive to the choice of this hyper-parameter, as evidenced by the steeper slopes of the curves in Figure 10. In contrast, with QS no kernel-type needs to be selected and no hyper-parameter tuning seems to be necessary, regardless of distribution shape and sample size, and sensitivity of the entropy estimate to precise choice of the number of quantiles   is relatively small even for smaller sample sizes (see Figure 4).

  • Paragraph 56 in the “Discussion and Conclusions” section has been modified as follows

[New Para 56] In principle, the QS approach provides a relatively simple method for obtaining accurate estimates of entropy from data samples, along with an idea of the estimation-uncertainty associated with sampling variability. It appears to have an advantage over BC and KD since the most important hyper-parameter to be specified, the number of quantiles , does not need to be tuned and can apparently be set to a fixed fraction () of the sample size, regardless of pdf shape or sample size.  In contrast, for BC the optimal number of bins  varies with pdf shape and sample size and, since the underlying pdf shape is usually not known beforehand, it can be difficult to come up with a general rule for how to accurately specify this value.  Similarly, for KD, without prior knowledge of the underlying pdf shape (and especially when the pdf may be non-smooth) it can be difficult to know what kernel-type and hyper-parameter settings to use.  Therefore, QS is both easier to apply (no hyper-parameter tuning required) and potentially more accurate than BC or KD when applied to data from an unknown distributional form, particularly since the the piecewise linear interpolation between CDF points makes it applicable to pdfs of any arbitrary shape, including those with sharp discontinuities in slope and/or magnitude.

  • The “Acknowledgements: section has been modified as follows

We are grateful to the two anonymous reviewers who provided comments and suggestions that helped to improve this paper, and especially to reviewer # 2 who insisted that we provide a comparison with the Parzen-window kernel density method. We also acknowledge the many members of the GeoInfoTheory community (https://geoinfotheory.org) who have provided both moral support and extensive discussion of matters related to Information Theory and its applications to the Geosciences; without their existence and enthusiastic engagement it is unlikely that the ideas leading to this manuscript would have occurred to us. The first author acknowledges partial support by the Australian Research Council Centre of Excellence for Climate Extremes (CE170100023). The QS and BC algorithms used in this work are freely accessible for non-commercial use at https://github.com/rehsani/Entropy. The KD algorithm used in this work is downloadable from https://webee.technion.ac.il/~yoav/research/blind-separation.html.

  • The following additional References have been Cited

Schwartz S, Zibulevsky M and YY Schechner (2005), Fast kernel entropy estimation and optimization, Signal Processing 85, pp. 1045–1058

Viola PA (1995), Alignment by Maximization of Mutual Information, PhD Thesis, Massachusetts Institute of Technology – Artificial Intelligence Laboratory.

Reviewer 2 Report

I appreciate the response to my request for comparison to other methods, but I have to disagree with the assessment that the hyperparameter tuning is cumbersome and difficult to do.  Pseudolikelihood methods for Parzen window estimates are actually incredibly easy to implement and quite effective.  I must insist on comparison to some of these other methods that have hyperparameters.  I am fine with a briefly-described result that says that this new method is worse due to the fact that it doesn't have as much to tune, but that should be known.  I would actually say that you have a hyperparameter, which is how many quantiles are used, and so are on par with, say, Parzen window estimates for difficulty.

Author Response

(The authors gave the same response as above.)

Round 3

Reviewer 2 Report

I really appreciate that you added an analysis of KD methods-- I will ask for one more graph and then leave it at that.  I think there should be a final graph that shows the percentage bias in the optimal entropy estimate (where you have optimized hyperparameters for each method somehow that doesn't directly minimize bias, which requires ground truth knowledge) as a function of N_S for all methods, to really see the comparison.

Author Response

Reviewer #2:

Comment: I really appreciate that you added an analysis of KD methods-- I will ask for one more graph and then leave it at that.  I think there should be a final graph that shows the percentage bias in the optimal entropy estimate (where you have optimized hyperparameters for each method somehow that doesn't directly minimize bias, which requires ground truth knowledge) as a function of N_S for all methods, to really see the comparison.

Response: Thanks for this really excellent suggestion, that we should have thought of ourselves J.

The revised paper now includes a new figure (Figure 12) that includes this comparison.  For the QS method we use a fixed value for the hyperparameter (Number of quantiles equal to 25% of the sample size), so no tuning is performed.  For both KD and BC we optimize the corresponding hyperparameter (kernel std for KD, and bin width for BC) to maximize the Likelihood of the data (minimize the cross entropy of the estimated density and the data), and do this separately for each time we draw a sample from the parent density, then use the corresponding “optimal” estimate of the pdf to compute the entropy.  For all three methods we run 500 replicates for each sample size for each of the test pdfs, re-drawing samples directly form the parent pdf, to examine the sampling variability.  The boxplot results indicate that QS is relatively unbiased, with expected bias close to 1% regardless of sample size or pdf type, even for small samples (e.g. 100).  For KD and BC, the bias increases (negatively) as sample size is decreased. BC is the worst, having the largest negative expected bias of approximately  for sample sizes of  whereas, as expected, KD is better and has negative expected bias pdf of about  for sample sizes of .

Proposed Revision: In light of these findings, we propose making the following changes to the manuscript to include the new figure, and to illustrate the relative properties of the QS, BC and KD methods (new text shown in red).

  • The Abstract has been modified to read as follows:

[New Abstract] We develop a simple Quantile Spacing (QS) method for accurate probabilistic estimation of one-dimensional entropy from equiprobable random samples, and compare it with the popular Bin-Counting (BC) and Kernel Density (KD) methods. In contrast to BC, which uses equal-width bins with varying probability mass, the QS method uses estimates of the quantiles that divide the support of the data generating probability density function (pdf) into equal-probability-mass intervals. And, whereas BC and KD each require optimal tuning of a hyper-parameter whose value varies with sample size and shape of the pdf, QS only requires specification of the number of quantiles to be used.

Results indicate, for the class of distributions tested, that the optimal number of quantiles is a fixed fraction of the sample size (empirically determined to be ), and that this value is relatively insensitive to distributional form or sample size. This provides a clear advantage over BC and KD since hyper-parameter tuning is not required. Further, unlike KD, there is no need to select an appropriate kernel-type, and so QS is applicable to pdfs of arbitrary shape, including those with discontinuous slope and/or magnitude.

Bootstrapping is used to approximate the sampling variability distribution of the resulting entropy estimate, and is shown to accurately reflect the true uncertainty. For the four distributional forms studied (Gaussian, Log-Normal, Exponential and Bimodal Gaussian Mixture), expected estimation bias is less than 1% and uncertainty is low even for samples of as few as  data points; in contrast, for KD the small sample bias can be as large as  and for BC as large as . We speculate that estimating quantile locations, rather than bin-probabilities, results in more efficient use of the information in the data to approximate the underlying shape of an unknown data generating pdf.

  • Paragraph 56 has been modified as follows (now becomes paragraphs 56 and 57):

[New Para 56 and 57] The QS approach provides a relatively simple method for obtaining accurate estimates of entropy from data samples, along with an idea of the estimation-uncertainty associated with sampling variability. It appears to have an advantage over BC and KD since the most important hyper-parameter to be specified, the number of quantiles , does not need to be tuned and can apparently be set to a fixed fraction () of the sample size, regardless of pdf shape or sample size.  In contrast, for BC the optimal number of bins  varies with pdf shape and sample size and, since the underlying pdf shape is usually not known beforehand, it can be difficult to come up with a general rule for how to accurately specify this value.  Similarly, for KD, without prior knowledge of the underlying pdf shape (and especially when the pdf may be non-smooth) it can be difficult to know what kernel-type and hyper-parameter settings to use. 

Besides being simpler to apply, the QS approach appears to provide a more accurate estimate of the underlying data generating pdf than either BC or KD, particularly for smaller sample sizes. This is illustrated clearly by Figure 12 where the expected percent entropy estimation error is plotted as a function of sample size  for each of the three methods. For QS, the fractional number of bins was fixed at  regardless of pdf form or sample size; in other words, no hyperparameter tuning was performed.  For each of the other methods, the corresponding hyperparameter (kernel standard deviation  for KD, and bin width ∆ for BC) was optimized for each random sample, by finding the value that maximizes the Likelihood of the sample. As can clearly be seen, the QS-based estimates remain relatively unbiased even for samples as small as 100 data points, whereas the KD- and BC-based estimates tend to get progressively worse (negatively biased) as sample sizes are decreased. Overall, QS is both easier to apply (no hyper-parameter tuning required) and likely to be more accurate than BC or KD when applied to data from an unknown distributional form, particularly since the the piecewise linear interpolation between CDF points makes it applicable to pdfs of any arbitrary shape, including those with sharp discontinuities in slope and/or magnitude. A follow-up study investigating the accuracy of these methods when faced with data drawn from complex, arbitrarily shaped, pdfs is currently in progress and will be reported in due course.

  • We have added a new Figure 12 as follows:

Figure 12: Plots showing expected percent error in the QS- (blue), KD- (purple) and BC-based (green) estimates of entropy derived from random samples, as a function of sample size ; box plots are shown side by side to improve legibility. Results are averaged over  trials obtained by drawing sample sets of size  from the theoretical pdf, where  and  are set to be the smallest and largest data values in the particular sample. For QS, the fractional number of bins was fixed at  regardless of pdf form or sample size.  For KD and BC, the corresponding hyperparameter (kernel standard deviation  and bin width ∆ respectively) was optimized for each random sample by finding the value that maximizes the Likelihood of the sample. Results show clearly that QS-based estimates are relatively unbiased, even for small sample sizes, whereas KD- and BC-based estimates can have significant negative bias when sample sizes are small.

[Figure is reproduced on the next page]
